# SheAttack: A Silhouette Score Motivated Restricted Black-box Attack on Graphs

## Abstract

Graph Neural Networks (GNNs) have gained large popularity in various applications, with their vulnerability against adversarial attacks also being brought up. Despite the numerous graph attacks proposed, few have focused on the Restrict Black-box attack, where attackers only have access to node features and the graph structure. Existing works in this setting aim to perform destructive attacks by degrading the quality of victim graphs yet imposing the homophily assumption or requiring high computational complexity. To address these challenges, we propose the Modified Silhouette Score (MSS) as a measure of a graph's quality, and demonstrate its generalizability across graphs of different homophily levels through theoretical analysis. Using MSS as the objective, we present SheAttack, an efficient attack that effectively reduces the distinguishability of nodes. We conduct experiments on both synthetic and real-world graphs to validate the effectiveness of SheAttack in both homophilic and heterophilic settings. We find that even without prior knowledge of labels or the victim model, our method shows comparable performance to split-unknown white-box attacks.

## 1 Introduction

Graph Neural Networks (GNNs) have emerged as a powerful tool in various graph learning tasks (Hu et al., 2020) and found applications in a wide range of fields (Gasteiger et al., 2021; Wu et al., 2023; Senior et al., 2023). However, recent studies have revealed that GNNs are also vulnerable to adversarial attacks (Dai et al., 2018; Zügner et al., 2018; Zügner & Günnemann, 2019). Unlike most attacks that target features in vision and language data, graph-specific attacks enable structural modifications that add or delete edges from the graph, which can have a devastating impact on GNNs' performance. In node classification, such structural perturbations can even cause GNNs to perform worse than simple Multilayer Perceptrons (MLPs), which use only node features as inputs (Zügner & Günnemann, 2019; Xu et al., 2019).

The vulnerability of GNNs has sparked significant interest in developing structural attacks under different settings. Generally, attacks on graphs can be classified into white-box, grey-box, and black-box attacks based on the available information to attackers, in descending order of knowledge (Jin et al., 2020). The powerful white-box and grey-box attacks have access to training/test splits but are overly dependent on this split information. When the training/test sets are resplit, these attacks only show a marginal advantage over black-box attacks (Zhan & Pei, 2021). Furthermore, the splits, node labels, and models are often defined by defenders according to their personal needs. For example, in a social media friendship network, users' properties and relations are public, but the downstream tasks and models used are up to defenders' preferences. To design more general attacks that poison the whole graph, the Restrict Black-box attack (RBA) setting is proposed (Chang et al., 2020), where only training inputs excluding node labels, are known to attackers. Although several attempts at designing RBAs have been made, existing methods often rely on prior assumptions, such as the low-rank property of node embeddings (Chang et al., 2020) or the homophily assumption of the graph (Li et al., 2022). While research on heterophilic graphs has received increasing attention, current RBAs are not guaranteed to maintain desirable performance when transferred to heterophilic graphs. Currently, it is still unknown how to design an RBA that is applicable to heterophilic graphs.

To shed light on the limitations of current RBAs against heterophily, we take a step back and look at how GNNs work. In node classification, GNNs excel at leveraging message passing to denoise

raw features. On informative graphs, GNNs push node representations of different classes to stay far away from each other. Recent studies have found that, as long as the graph structure is predictive for node labels, GNNs successfully achieve the goal on graphs either of high homophily or high heterophily Ma et al. (2022b). Since GNNs do not rely on graph homophily to have outstanding performance, current RBAs, which primarily try to reduce the similarity between connected nodes, fail to cover all the cases.

Although GNNs can handle both homophilic and heterophilic graphs, if node embeddings of different classes fail to be separated after message-passing, GNNs will generally underperform. For example, when the over-smoothing phenomenon occurs, even GNNs with heterophilic designs tend to fail Yan et al. (2022). Therefore, it would be wiser to consider the closeness between node embeddings as the objective of the attack. To quantify the separation of node embeddings, we introduce the Silhouette Score, which is typically used to measure intra-class and inter-class distances in clustering tasks where a higher score indicates better clustering. By lowering the score for propagated embeddings, our approach poses a substantial challenge to all GNNs. We further enhance the Silhouette Score to make it more robust and applicable in the context of attacks. Based on the enhanced score, we present SheAttack and its approximate version, which effectively push inter-class node embeddings closer. During optimization, we employ Greedy Randomized Block Coordinate Descent, ensuring the scalability of SheAttack in both time and space. Our contribution can be summarized as follows:

1. We investigate graph structural attacks under the RBA setting and highlight the limitations of previous methods in terms of generalizability to heterophilic settings.

2. We introduce a novel restricted black-box attack framework, SheAttack and its scalable version, which attacks the graph structure by leveraging distances between node embeddings.

3. We provide a theoretical analysis of the change in class-wise distances of SheAttack, demonstrating its ability in both homophilic and heterophilic settings.

4. We conduct extensive experiments on diverse datasets, verifying the effectiveness of SheAttack. Notably, we compare the performance of SheAttack with shuffled white-box attacks, demonstrating their comparable performance and the necessity of studying the RBA setting.

## 2 PRELIMINARIES

**Notations.** Denote an undirected graph as $G = (V, E)$, where $V$ is the node set with $|V| = n$, and $E$ is the edge set with $|E| = m$. The graph structure can also be represented by adjacency matrix $\mathbf{A} \in \{0, 1\}^{n \times n}$. The node degree matrix is a diagonal matrix defined as $\mathbf{D} = \text{diag}\{d_1, \cdots, d_n\}$, where $d_j = \sum_i \mathbf{A}_{ij}$. Let $\hat{\mathbf{A}} = \mathbf{D}^{-1/2} \mathbf{A} \mathbf{D}^{-1/2}$ be the normalized adjacency matrix. Similarly, we define the normalized adjacency matrix with self-loops as $\tilde{\mathbf{A}}$. In node classification tasks, nodes are assigned node features and node labels. We denote node features as $\mathbf{X} \in \mathbb{R}^{n \times d}$, and node labels as $\mathbf{Y} \in \{0, 1, \ldots, c-1\}^n$, where $c$ is the number of classes.

**GNN architechture.** Given node features $\mathbf{X}$ and adjacency matrix $\mathbf{A}$, we denote a GNN model with parameters $\theta$ as $f_\theta(\mathbf{X}; \mathbf{A})$. A GNN model usually comprises multiple GNN layers and follows a message-passing scheme. In each layer, node representations $\mathbf{h}_i^{(l)} \in \mathbb{R}^{h_l}$ are recursively updated as:

$$\mathbf{h}_i^{(l)} = \sigma \left( \text{AGGR} \left( \alpha_{ij} \mathbf{h}_j^{(l-1)} \mathbf{W}^{(l)} \mid j \in \mathcal{N}(i) \cup \{i\} \right) \right),$$

where $\mathcal{N}(i)$ is the set of node $i$'s neighbors, $\mathbf{W}^{(l)} \in \mathbb{R}^{h_{l-1} \times h_l}$ is the linear transformation, $\sigma$ is the activation function, and AGGR is the aggregation function, e.g., SUM or MEAN. The term $\alpha_{ij}$ is a normalization coefficient, which is set as $\alpha_{ij} = 1/\sqrt{d_i d_j}$ in Graph Convolutional Network (GCN) Kipf & Welling (2017) and calculated using attention in Graph Attentional Network (GAT) Velickovic et al. (2018). The output embeddings can be later used in downstream tasks.

**Problem definition.** In this paper, we focus on **non-targeted graph structural attacks**. These attacks aim to decrease the prediction accuracy of all test nodes by modifying the graph structure. The graph structure can only be perturbed within a limited budget, denoted as $\Delta$, to make the attack unnoticeable. Formally, given loss function $\mathcal{L}$, the objective of the attacker is:

$$\max_{\mathbf{A}'} \mathcal{L}(f_\theta(\mathbf{X}, \mathbf{A}')), \quad \text{s.t.} \quad \|\mathbf{A}' - \mathbf{A}\|_0 \leq \Delta,$$

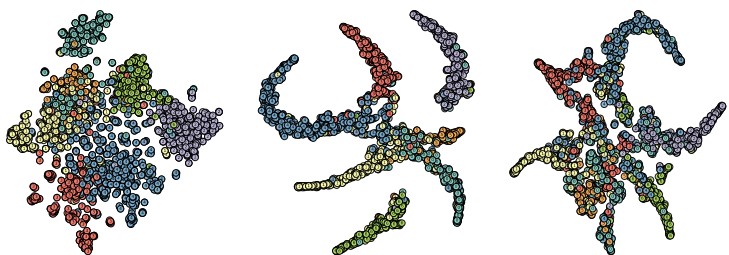

Figure 1: The t-SNE visualization of raw features (left), GCN outputs on the clean graph (middle), and GCN outputs on the perturbed graph (right) in dataset Cora-ML.

where $\mathbf{A}'$ is the perturbed adjacency matrix. If $\theta$ is trained before perturbations, the attacks are referred to as poison attacks and otherwise evasion attacks. Although we do not focus on the time of occurrence of RBA, we transfer the poisoned graph to both settings during evaluation. Based on the attacker's knowledge, graph structural attacks can be classified into white-box Xu et al. (2019); Geisler et al. (2021), grey-box Zügner & Günnemann (2019), and black-box attacks Dai et al. (2018); Chang et al. (2020); Lin et al. (2022). Among them, the RBA setting is the most strict, in which only node features and the graph structure are available for the attackers.

## 3 MEASURING DIFFICULTY OF NODE CLASSIFICATION

To effectively carry out black-box attacks, it is crucial for attackers to establish a clear definition of a successful attack. When node labels and victim models are known, the quality of graphs can be easily determined using the victim models' accuracy on the test set. However, the task becomes nontrivial when labels and victim models are unavailable. Previous RBAs address this problem by imposing prior assumptions, like the low-rank Bojchevski & Günnemann (2019); Chang et al. (2020) property of node embeddings or homophily assumption Li et al. (2022), yet limits the applicability of RBAs to different tasks. Therefore, we raise the question: *"Can we create a challenging classification task that targets all classifiers on all graphs?"*

### 3.1 SILHOUETTE SCORE

In node classification, GNNs outperform previous methods by utilizing message passing to denoise node embeddings. Ideally, GNNs separate node embeddings of different classes to stay far away from each other in the embedding space. However, if this process is hindered, the performance of GNNs will suffer from significant degradation. In Figure 1, we present the t-SNE Van der Maaten & Hinton (2008) visualization of raw features and node embeddings generated by GCN on the clean and perturbed graphs. On clean graphs, GCN obtains node embeddings that are well separated for different classes. Yet, on the poisoned graph, node embeddings of all classes get suppressed closer, which makes GCN lose its expressiveness in classifying nodes. The significant differences in the two scenarios inspired us to use distances between nodes to reflect the quality of the graph.

To formally measure the proximity of node embeddings, we introduce Silhouette Score Rousseeuw (1987). Silhouette Score captures distances between intra-class and inter-class instances to quantify the difficulty of a clustering problem. In node clustering, denote the cluster of nodes as $C_I = \{i \mid y_i = I\}$. The average intra-class distance and smallest average inter-class distance for node $i$ in cluster $I$ are:

$$a_i = \frac{1}{|C_I| - 1} \sum_{j \in C_I, i \neq j} \mathcal{D}(i, j), \quad b_i = \min_{J \neq I} \frac{1}{|C_J|} \sum_{j \in C_J} \mathcal{D}(i, j). \tag{1}$$

Here $\mathcal{D}(i, j)$ is a distance metric, which can be calculated using the embeddings of node $i$ and $j$. Then, Silhouette Scores for node $i$ and graph $G$ are

$$s_i = \frac{b_i - a_i}{\max(b_i, a_i)}, \quad s_G = \frac{1}{n} \sum_{v_i \in V} s_i.$$

By definition, Silhouette Score lies in the range of $[-1, 1]$. A larger Silhouette Score represents a larger relative inter-class distance and an easier node clustering problem.

## 3.2 MODIFICATIONS TO SILHOUETTE SCORE

To attack a graph, a straightforward idea is to minimize its Silhouette Score. However, we have several concerns if directly using it as the objective in attacks.

**Quality of clusters.** Since node labels are inaccessible, we must first conduct node clustering to define the Silhouette Score. Because clustering results are hard to be ideal in all cases, we risk mistakenly pushing the attacked node away from nodes in different classes. To provide a safeguard for this situation, we introduce **Shift Loss** as a robust alternative for intra-class distance $a_i$, which is defined as: $\mathcal{L}_{\text{shift}} = \|\mathbf{h}_i' - \mathbf{h}_i\|$. Here, $\mathbf{h}_i'$ is the embedding of $i$ after the attack. Shift loss makes sense if we consider node $i$ itself as the representative of its cluster. By maximizing Shift, we can robustly create a larger intra-class distance and therefore reduce its Silhouette Score.

**Limit of scope.** For the original Silhouette Score in Equation equation 1, minimum inter-class distance is used to describe the closeness of a node to other clusters. However, this metric may be too strict as the objective for attacks. Generally, a node becomes more indistinguishable as long as its distance to nodes in other clusters is shortened. However, when using the minimum inter-class distance as the objective, only nodes in its nearest neighboring cluster are taken into account. If we choose perturbations based on the gradients of the Silhouette Score with respect to edge flips, distant clusters are disregarded. For example, in Figure 2, the embedding of the attacked node is shifted under attacks. Shifts towards either red nodes or green nodes decrease the distinguishability of the attacked node. However, when using the original Silhouette Score, the inter-class distance of the attacked node is influenced only by red nodes, which are closer to the blue cluster. To expand the scope of attackers, we consider average inter-class distances instead of the minimum in Equation equation 1:

$$\hat{b}_i = \frac{1}{n - |C_I|} \sum_{j \in C_J, J \neq I} \mathcal{D}(i, j).$$

Here, we compute the average distance of node $v_i$ to nodes in other clusters. We refer to the Silhouette Score calculated with $\hat{b}_i$ as the **Modified Silhouette Score (MSS)**. This metric puts more possibly harmful choices into consideration during attacks and is also effective in measuring the quality of a graph. In Figure 3, we plot the change in MSS, which is calculated using clusters from Kmeans and ground-truth labels, under a greedy white-box attack GRBCD Geisler et al. (2021) and the random attack. GRBCD greedily adds/deletes the most influential edges to/from the graph to decrease the performance of victim models. From the figure, we see that the attack adaptively flips edges that lower the MSS of the graph. Compared with Random Attack, which randomly flips edges on the graph, GRBCD results in a larger drop in MSS.

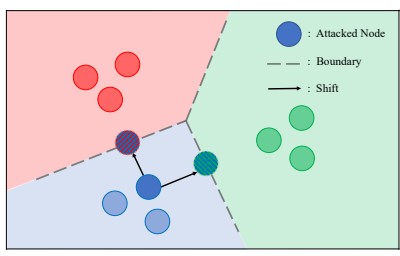

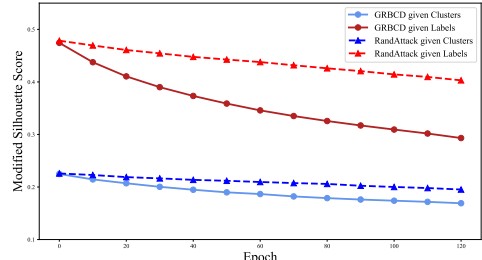

Figure 2: Shift of node embedding under attacks. Mixing the attacked node with either red or green nodes makes it indistinguishable.

Figure 3: Change of MSS under GRBCD and Random Attack with respect to epoch. GRBCD results in a larger change in MSS than Random Attack.

## 4 SHEATTACK

In this section, we provide the complete algorithm of **SheAttack** (Silhouette Score Based Attack).

### 4.1 THE ALGORITHM

Given node attributes $\mathbf{X}$ and adjacency matrix, we first conduct Principal Component Analysis (PCA) to raw features and obtain the projected node features $\mathbf{X}_{\text{proj}} \in \mathbb{R}^{n \times p}$. Then, we incorporate graph structure information by computing message propagation on graphs. Similar to GCN, we propagate projected features twice on the normalized adjacency matrix with self-loops, obtaining $\tilde{\mathbf{X}} = \tilde{\mathbf{A}}^2 \mathbf{X}_{\text{proj}}$. To obtain node clusters, we apply Kmeans (Lloyd, 1982) with Kmeans++ seeding (Arthur & Vassilvitskii, 2007) to $\tilde{\mathbf{X}}$. Denote the number of clusters as $k$, the algorithm returns node clusters $\hat{C} = \{\hat{C}_0, \hat{C}_1, \cdots, \hat{C}_{k-1}\}$ and corresponding centroids. (We use $\hat{C}_i$ for clusters generated by clustering algorithms and $C_i$ for ground-truth classes). After that, we calculate the intra/inter-class distances as

$$a_i' = \frac{1}{\left|\hat{C}_I\right| - 1} \sum_{j \in \hat{C}_I, i \neq j} \mathcal{D}(i,j), \quad b_i' = \frac{1}{n - \left|\hat{C}_I\right|} \sum_{j \in \hat{C}_J, J \neq I} \mathcal{D}(i,j).$$

Here, $\mathcal{D}(i,j)$ is a distance metric calculating the distance between $\tilde{\mathbf{X}}_i$ and $\tilde{\mathbf{X}}_j$. The Silhouette Score-based loss and its combination with shift loss are:

$$\mathcal{L}_{\text{she}} = -s_G = -\frac{1}{n} \sum_{v_i \in V} \frac{b_i' - a_i'}{\max\left(b_i', a_i'\right)}, \quad \mathcal{L}_{\text{atk}} = \mathcal{L}_{\text{she}} + \lambda \mathcal{L}_{\text{shift}}, \tag{2}$$

where $\lambda \geq 0$ is a hyperparameter. Denote the perturbed graph structure as $\mathbf{A}'$, the inputs of Shift Loss are $\tilde{\mathbf{X}}$ calculated using $\mathbf{A}$ and $\mathbf{A}'$ during attacks. The objective can then be written as $\mathcal{L}_{atk}(\mathbf{X}, \mathbf{A}', \hat{C})$.

After defining the loss term, we use the Greedy Randomized Block Coordinate Descent for optimization (Geisler et al., 2021). We adopt this optimization technique for its linear time and space complexity w.r.t the block size while enjoying a desirable performance. Specifically, during each epoch, we calculate the gradients of the objective function w.r.t possible edge flips in a randomly sampled block. We conduct the top-K edge flips with the largest gradients to obtain the modified adjacency matrix $\mathbf{A}'$, and repeat the process until a given budget is met. The pseudocode of our algorithm is given in the Appendix B.

### 4.2 APPROXIMATION TO SCALE UP

The procedure of SheAttack comprises PCA, Kmeans, loss computation, and optimization, whose corresponding time complexities are $O(nd^2 + d^3)$, $O(npk)$, $O(n^2)$, and $O(b)$ ($b$ is the block size linear to the number of edges), respectively. On large graphs, the bottleneck of SheAttack in time complexity is $O(n^2)$ in computing node-wise distances. To improve the scalability, we introduce an approximate version that computes distances between nodes and cluster centroids. Let the $k$ centroids returned by clustering be $\{\boldsymbol{\mu}_1, \cdots, \boldsymbol{\mu}_k\}$, where each $\boldsymbol{\mu}_i$ stands for the center of cluster $I$. The node-wise intra-class and inter-class distances for node $i$ of cluster $I$ are:

$$\hat{a}_i = \mathcal{D}(i, \mu_I), \quad \hat{b}_i = \frac{1}{k-1} \sum_{J \neq I} \mathcal{D}(i, \mu_J).$$

By substituting $\hat{a}_i$ and $\hat{b}_i$ into loss equation 2, we obtain an approximate version of the loss. The time complexity in computing distances is now $O(nk)$, which is highly scalable. All operations can be done using the sparse matrix, so the space complexity of SheAttack is $O(b)$ as well.

### 4.3 INTERPRETATIONS OF SHEATTACK

In this section, we study the behavior of SheAttack on graphs generated by the Contextual Stochastic block model (cSBM) (Deshpande et al., 2018). As a classical graph generation model, cSBM and its variants are widely used for analyzing graph clustering (Fortunato & Hric, 2016) and GNNs (Chien et al., 2021; Ma et al., 2022b). Here, we consider a two-class node classification problem with classes written as $c_1$ and $c_2$. Edges are formed between intra-class nodes with probability $p$ and between inter-class nodes with probability $q$. The node feature $\mathbf{x}_i$ in class $c_i$ follows Gaussian distribution $N(\boldsymbol{\mu}_i, \mathbf{I})$, where $\boldsymbol{\mu}_i \in \mathbb{R}^d$ is the class centroid. Similar to (Ma et al., 2022b), we focus on the message passing and disregard the linear transformation. The embedding of a node $v_i$ is $\mathbf{h}_i = \frac{1}{d_i} \sum_{j \in \mathcal{N}(i)} \mathbf{x}_j$,

where $d_i$ is the node degree. Suppose node $v_i$ belongs to class $c_1$. Denote intra-class perturbations as $\Delta_1$ and inter-class perturbations $\Delta_2$. A positive $\Delta_1$ ($\Delta_2$) stands for adding edges and otherwise deleting. Denote the propagated embedding after perturbations as $\mathbf{h}'_i$, then

$$\mathbf{h}_i \sim N\left(\frac{p\boldsymbol{\mu}_1 + q\boldsymbol{\mu}_2}{p+q}, \frac{1}{\sqrt{d_i}}\mathbf{I}\right), \mathbf{h}'_i \sim N\left(\frac{(pd_i + \Delta_1)\boldsymbol{\mu}_1 + (qd_i + \Delta_2)\boldsymbol{\mu}_2}{pd_i + qd_i + \Delta_1 + \Delta_2}, \frac{1}{\sqrt{d_i + \Delta_1 + \Delta_2}}\mathbf{I}\right).$$

**Theorem 4.1.** *(Ma et al., 2022b)* (**Informal**) *For any node $i$ in the cSBM graph, the linear classifier defined by the decision boundary $\mathcal{P} = \left\{\mathbf{x} \mid \mathbf{w}^\top \mathbf{x} - \mathbf{w}^\top (\boldsymbol{\mu}_1 + \boldsymbol{\mu}_2)/2\right\}$ has a lower probability of misclassifying $\mathbf{h}_i$ than $\mathbf{x}_i$ when $d > (p+q)^2/(p-q)^2$.*

The above theorem clarifies when a linear classifier has better distinguishability. On graphs of high homophily or heterophily, node $v_i$ has a higher probability of being correctly classified. In other words, homophily loss (Li et al., 2022) which aims to lower the homophily level, could fail to improve attack performance if the victim graph is heterophilic. In contrast, the objective of SheAttack is able to handle both cases. Denote the propagated centroids of the two classes as $\boldsymbol{\mu}'_1 = \frac{p\boldsymbol{\mu}_1 + q\boldsymbol{\mu}_2}{p+q}$ and $\boldsymbol{\mu}'_2 = \frac{p\boldsymbol{\mu}_2 + q\boldsymbol{\mu}_1}{p+q}$. The intra-class and inter-class distances satisfy the following theorem:

**Theorem 4.2.** *Let $a_i = \|\mathbf{h}'_i - \boldsymbol{\mu}'_1\|^2$ be the square of intra-class distance under perturbation, $b_i = \|\mathbf{h}'_i - \boldsymbol{\mu}'_2\|^2$ be the square of inter-class distance under perturbation, then*

$$\mathbb{E}[a_i] = \left(\frac{\Delta_1 q - \Delta_2 p}{(p+q)(pd_i + qd_i + \Delta_1 + \Delta_2)}\right)^2 \|\boldsymbol{\mu}_1 - \boldsymbol{\mu}_2\|^2 + c,$$

$$\mathbb{E}[b_i] = \left(\frac{pm_p - qm_q}{(p+q)(pd_i + qd_i + \Delta_1 + \Delta_2)}\right)^2 \|\boldsymbol{\mu}_1 - \boldsymbol{\mu}_2\|^2 + c,$$

*where $c = \mathrm{Tr}\left(\mathrm{Cov}\left(\mathbf{h}'_i\right)\right) = \frac{d}{\sqrt{d_i + \Delta_1 + \Delta_2}}$, and $m_p = pd_i + \Delta_1$ and $m_q = qd_i + \Delta_2$ are the homophilic and heterophilic edges after perturbations.*

The proof and more explanation are included in Appendix D.1. To minimize $b_i$, SheAttack would prefer to enlarge $m_q$ if $p > q$, or enlarge $m_p$ if $p < q$, which results in a tougher classification task as shown in Theorem 4.1. Note that shift loss which only focuses on maximizing $a_i$ fails to fit in both settings. On heterophilic graphs where $q > p$, a negative $\Delta_1$ would result in larger shifts while the homophily level goes up at the same time. This phenomenon occurs as the direction of shift is not provided, and node embeddings could shift to a position far away from other classes. So in practice, SheAttack combines both objectives to ensure generalizability and effectiveness.

## 5 EXPERIMENT

### 5.1 SETTINGS

**Datasets.** We generate cSBM graphs following the setting in (Chien et al., 2021). For small real-world datasets, we adopt homophilic graphs Cora-ML (McCallum et al., 2000), Citeseer, and PubMed (Sen et al., 2008), and heterophilic datasets Chameleon, Squirrel (Pei et al., 2020; Rozemberczki et al., 2021), and Roman-Empire (Platonov et al., 2023). We transfer graphs into undirected and select the largest connected components on small real-world datasets following (Zügner & Günnemann, 2019). We also include large datasets ogbn-arxiv and ogbn-products (Hu et al., 2020) with fixed split to test the efficiency of attacks. Details of datasets and splits are given in the Appendix C.2.

**Baselines. Random**: Randomly add or delete edges on the graph; **DICE** (Waniek et al., 2018): Randomly add edges between inter-class nodes or delete edges between intra-class nodes. Note that DICE is **not** a strict black-box attack, and we introduce a hyperparameter threshold for Random and DICE attacks that controls the proportion of add and delete operations for enhancement. **GFAttack** (Chang et al., 2020): A targeted RBA. We modify it into an untargeted attack by transferring the objective to the untargeted setting. **SPAC** (Lin et al., 2022): A loss-based RBA and its approximated variant that maximizes the spectral distance during perturbation. **SelfAttack**: A self-attack algorithm that adopts a two-layer GCN as the surrogate similar to (Zhan & Pei, 2021). The surrogate model is trained using clusters given by Kmeans. **PEEGA** (Li et al., 2022): An RBA that combines shift loss and homophily loss as the objective. The variant without "-Comb" means only homophily loss is included. For SPAC

Table 1: Performance of GCN on perturbed cSBM graphs with perturb ratio 0.20. The datasets are named cSBM($\Phi * 100$). The best results are **bold**, and the second-best results are underlined.

| Attack | cSBM+25 Evasion | cSBM+25 Poisoning | cSBM-25 Evasion | cSBM-25 Poisoning | cSBM+50 Evasion | cSBM+50 Poisoning | cSBM-50 Evasion | cSBM-50 Poisoning |
|---|---|---|---|---|---|---|---|---|
| Clean | 81.25±0.32 | | 57.42±1.00 | | 93.39±0.18 | | 74.11±0.30 | |
| Random | 79.68±0.41 | 78.99±0.78 | 56.74±0.83 | 56.34±0.65 | 91.09±0.21 | 90.51±0.54 | 71.46±1.02 | 70.18±0.71 |
| DICE | 72.51±0.93 | 69.06±0.67 | 63.33±1.39 | 68.88±0.71 | **84.38±1.32** | **82.10±1.30** | 83.07±0.53 | 84.93±0.69 |
| SPAC-A | 80.59±0.27 | 80.64±0.21 | 57.66±0.73 | 59.25±0.89 | 93.15±0.47 | 92.90±0.56 | 76.35±0.44 | 76.48±0.30 |
| GFAttack | 79.84±0.26 | 79.28±0.53 | 57.97±1.13 | 58.43±1.61 | 91.92±0.38 | 91.46±0.36 | 73.81±1.25 | 73.39±0.87 |
| SelfAttack | 78.11±0.88 | 77.06±1.05 | 58.18±1.19 | 58.40±1.00 | 90.34±0.57 | 90.21±0.61 | 74.62±0.59 | 74.75±0.44 |
| PEEGA | 81.54±0.22 | 80.54±0.49 | 57.71±0.45 | 56.10±0.34 | 92.18±0.33 | 91.68±0.58 | 70.54±0.77 | 68.53±1.26 |
| PEEGA-Comb | 77.62±0.61 | 75.82±0.77 | 59.10±1.18 | 58.38±0.71 | 88.48±0.61 | 86.99±1.15 | 75.87±1.14 | 74.58±0.86 |
| She | **67.94±0.94** | 62.69±0.93 | **49.76±0.46** | 48.62±1.48 | 88.27±1.88 | 87.07±2.15 | **67.01±0.92** | **63.82±1.29** |
| She-Comb | 69.30±0.33 | 64.43±0.76 | 51.17±0.57 | 48.74±1.29 | 87.70±0.85 | 86.24±1.54 | 68.91±1.08 | 65.95±1.35 |
| She-A | 68.18±0.77 | **62.37±0.42** | 50.70±0.65 | 49.09±1.51 | 89.10±1.61 | 88.48±1.86 | 68.30±0.71 | 65.84±1.33 |
| She-A-Comb | 68.77±0.92 | 62.75±0.57 | 50.34±1.45 | 48.94±0.93 | 89.42±1.48 | 88.02±2.03 | 68.61±1.27 | 66.22±0.83 |

and GFAttack, we use PGD (Xu et al., 2019) for optimization following (Lin et al., 2022). Otherwise, methods are optimized using greedy block gradient descent. All approximated variants are attached with the suffix "-A", and the variants including shift loss are attached with the suffix "-Comb".

**Victim models.** We use a two-layer GCN as the victim model in evasion and poisoning settings. We also test the performance of attacks against other GNN variants and defense models in Appendix F.4 Full results, training details, and hyperparameters are left in the Appendix C.3.

## 5.2 EXPERIMENT RESULTS

**Results on cSBM graphs.** To validate the generalizability of SheAttack, we conduct experiments on graphs generated using cSBM. Similar to (Chien et al., 2021), we use $\Phi \in [-1, 1]$ to control the homophily of generated graphs, where a positive $\Phi$ homophily indicates a homophilic graph and vice versa. We use raw features for clustering in SheAttack as they are powerful enough in cSBM graphs. The results are summarized in Table 1.

We see DICE and PEEGA help GCN enjoy a performance improvement after the attack on heterophilic datasets, which verifies the restriction by relying on homophily assumption. Random attack is a relatively strong baseline on heterophilic graphs, given that it does not impose wrong prior knowledge. But it is not compatible with other attacks on homophilic graphs. Among all methods, SheAttack generally holds superiority.

**Results on real-world datasets.** We conduct untargeted structural black-box attacks on chosen datasets and test the performance of GCN under both evasion and Poisoning settings. We repeat experiments five times over different random seeds and splits without additional specifications. All results are reported using mean accuracy ± standard deviation. Results are summarized in Table 2, Table 3 and Table 4. SheAttack and its variants generally outperform other baselines with desirable efficiency to scale up.

**Ablation Study.** To better analyze the influential components of SheAttack, we conduct an ablation study about the objective in Table 5. Here "Shift" corresponds to shift loss as the objective, and "She-min" corresponds to the original Silhouette score as the objective. We see that the She loss and shift loss both contribute to the attack performance, while the unmodified Silhouette score performs worse because of its limited scope.

Table 4: Performance of GCN on ogbn-products with perturb ratio 0.10. Experiments are repeated once on the fixed split.

| Attack | Evasion | Poisoning |
|---|---|---|
| Clean | 75.80 | |
| Random | 68.67 | 70.29 |
| She-A | 63.84 | 67.40 |
| She-A-Comb | **62.93** | **67.28** |

## 6 RELATED WORKS

We classify black-box attacks into Candidate Node-based, Reinforcement learning-based (RL-based), and loss-based methods.

Table 2: Performance of GCN on homophilic graphs with perturb ratio 0.20. The best results are **bold**, and the second-best results are underlined. Entries marked with "*" take too much time.

| | Cora-ML | | CiteSeer | | PubMed | |
|---|---|---|---|---|---|---|
| Attack | Evasion | Poisoning | Evasion | Poisoning | Evasion | Poisoning |
| Clean | 85.94±0.74 | | 74.09±1.47 | | 86.59±0.26 | |
| Random | 81.95±1.27 | 81.73±1.14 | 71.50±1.43 | 69.80±1.03 | 82.57±0.33 | 83.03±0.22 |
| DICE | 79.63±1.34 | 79.72±0.97 | 69.37±1.31 | 66.91±1.72 | 77.70±0.46 | **78.42±0.14** |
| SPAC | 83.09±0.91 | 82.63±0.70 | 72.23±1.22 | 70.83±1.65 | * | * |
| SPAC-A | 82.55±1.05 | 82.80±0.77 | 72.80±1.74 | 72.35±1.54 | * | * |
| GFAttack | 82.28±1.09 | 82.06±1.30 | 72.07±1.72 | 71.40±1.74 | * | * |
| SelfAttack | 81.50±1.00 | 80.34±0.49 | 73.42±1.52 | 72.65±1.33 | 81.79±0.73 | 82.09±0.37 |
| PEEGA | 83.49±1.13 | 83.23±0.66 | 71.47±1.46 | 70.47±1.01 | 85.48±0.34 | 85.35±0.28 |
| PEEGA-Comb | 75.74±1.55 | 74.44±1.00 | 65.79±1.50 | 61.61±1.83 | 77.27±0.75 | 79.68±0.28 |
| She | 74.72±1.46 | 73.20±0.94 | 66.02±1.34 | 62.67±0.94 | 77.79±0.70 | 79.70±0.13 |
| She-Comb | 74.46±1.54 | 71.91±1.49 | 65.73±1.83 | 62.10±1.31 | **76.89±0.75** | 79.64±0.11 |
| She-A | 75.54±1.24 | 73.91±1.11 | 67.44±0.90 | 65.06±1.17 | 80.20±0.30 | 80.80±0.14 |
| She-A-Comb | **73.78±1.45** | **71.46±1.07** | **65.39±1.63** | **60.84±1.85** | 77.30±0.37 | 79.12±0.29 |

Table 3: Performance of GCN on heterophilic graphs with perturb ratio 0.20. The best results are **bold**, and the second-best results are underlined. Entries marked with "*" take too much time.

| | Chameleon | | Squirrel | | Roman-Empire | |
|---|---|---|---|---|---|---|
| Attack | Evasion | Poisoning | Evasion | Poisoning | Evasion | Poisoning |
| Clean | 67.59±2.48 | | 53.22±1.80 | | 49.85±0.54 | |
| Random | 59.65±0.81 | **56.45±1.17** | 38.42±2.23 | **35.02±2.12** | 49.64±0.55 | 50.38±0.32 |
| DICE | 60.26±1.37 | 58.99±1.37 | 41.65±1.30 | 44.40±2.15 | 48.74±0.74 | 49.74±0.40 |
| SPAC | 58.46±1.59 | 57.76±2.27 | 38.54±1.92 | 38.44±1.71 | * | * |
| SPAC-A | 60.04±1.07 | 61.45±2.13 | 38.33±1.39 | 41.23±1.24 | * | * |
| GFAttack | 61.84±2.22 | 60.31±2.91 | 40.83±0.53 | 42.17±1.12 | * | * |
| SelfAttack | 59.30±1.65 | 61.40±2.06 | 37.85±0.81 | 42.25±1.24 | 47.77±0.66 | 48.69±0.57 |
| PEEGA | 58.82±1.07 | 63.95±2.44 | 44.78±2.18 | 47.92±1.95 | 50.42±0.43 | 53.31±0.95 |
| PEEGA-Comb | 57.68±2.15 | 60.44±2.88 | 37.14±1.56 | 42.11±1.23 | 45.66±0.59 | 48.64±0.35 |
| She | 58.46±1.93 | 60.18±1.94 | 40.92±1.82 | 40.96±1.93 | 45.00±0.53 | **47.25±0.44** |
| She-Comb | 54.69±2.52 | 58.20±1.84 | 39.62±1.57 | 40.44±1.42 | **44.97±0.50** | 47.61±0.24 |
| She-A | 53.16±0.74 | 60.44±0.88 | **34.79±1.92** | 40.94±1.14 | 46.11±0.24 | 47.65±0.26 |
| She-A-Comb | **53.11±1.17** | 60.48±0.79 | 34.85±1.25 | 41.25±2.65 | 45.47±0.78 | 48.32±0.75 |

**Candidate node-based methods.** This line of work selects candidate nodes on the graph and perturbs their features. RWCS (Ma et al., 2020) chooses the candidate nodes based on random walks. The authors in InfMax (Ma et al., 2022a) formulate the problem as a misclassification maximization problem and connect it to a linear threshold Model in influence maximization. While also categorized into black-box attacks, these methods do not consider structural modifications.

**RL-based methods.** Reinforcement learning was first applied to black-box graph adversarial attacks in RL-S2V (Dai et al., 2018). The attack process is formulated using a Markov Decision Process (MDP). It focuses on targeted attacks and is applicable in both node-level and graph-level settings. ReWatt (Ma et al., 2021) is proposed as a rewiring strategy for graph-level untargeted attacks, which is more local and less noticeable. RL-based algorithms possess desirable theoretical complexity, but the output of victim models is required, and the running time is usually not ideal in practice.

**Loss-based methods.** Loss-based methods first define a target loss under the black-box setting and then view perturbation as parameters to be optimized. Several works propose loss terms in the spectral domain. Embedding attack (Bojchevski & Günnemann, 2019) first analyzes the vulnerability of graph embedding models, yet node attributes are not included. GF-attack (Chang et al., 2020) as a targeted attack assumes node embeddings to be low-ranked and maximizes the distance between node embeddings and their low-rank approximation. SPAC (Lin et al., 2022) considers a more straightforward objective function by maximizing spectral distance before and after perturbation. The above methods 1) require specific assumptions and 2) require eigendecomposition, which is of high computational complexity. PEEGA (Li et al., 2022) focuses on the shift loss and a regularized loss term minimizing the result homophily. Yet the space complexity is still $O(n^2)$, and the homophily assumption is required on victim graphs.

Table 5: Performance of GCN against attacks using different loss terms.

| | Cora-ML | | CiteSeer | | PubMed | |
|---|---|---|---|---|---|---|
| Attack | Evasion | Poisoning | Evasion | Poisoning | Evasion | Poisoning |
| Shift | 75.48±1.69 | 73.55±1.24 | 65.77±1.15 | **61.84±1.42** | 77.71±0.63 | 79.97±0.25 |
| She-min | 78.81±1.09 | 78.16±0.69 | 67.77±1.41 | 66.72±1.62 | 79.01±1.07 | 80.64±0.78 |
| She | 74.72±1.46 | 73.20±0.94 | 66.02±1.34 | 62.67±0.94 | 77.79±0.70 | 79.70±0.13 |
| She-Comb | **74.46±1.54** | **71.91±1.49** | **65.73±1.83** | 62.10±1.31 | **76.89±0.75** | **79.64±0.11** |
| | Chameleon | | Squirrel | | Roman-Empire | |
| Shift | 59.08±1.75 | 61.05±1.47 | **36.77±1.26** | 40.73±1.95 | 45.58±0.79 | 48.77±0.54 |
| She-min | 60.61±1.96 | 62.68±2.57 | 42.98±1.99 | 42.31±3.36 | 46.08±0.33 | 47.79±0.48 |
| She | 58.46±1.93 | 60.18±1.94 | 40.92±1.82 | 40.96±1.93 | 45.00±0.53 | **47.25±0.44** |
| She-Comb | **54.69±2.52** | **58.20±1.84** | 39.62±1.57 | **40.44±1.42** | **44.97±0.50** | 47.61±0.24 |

# 7 DISCUSSION & LIMITATION

**How strong is the white-box attack?** Comparing the experiment results of SheAttack with those under white-box attacks, a large performance gap still exists (Zügner & Günnemann, 2019; Xu et al., 2019). However, as mentioned by (Zhan & Pei, 2021; Li et al., 2023), the performance of white-box attacks heavily relies on the utilization of train/test splits. To erase the unfairness introduced by additional knowledge, we test white-box attacks GRPCD and PRBCD (Geisler et al., 2021) given shuffled splits. More concretely, we generate white-box attacks on a given split and transfer them to other splits. The results are shown in Table 6. Interestingly, SheAttack is now competitive to split shuffled white-box attacks, even if they are given ground-truth labels and model parameters. Based on this result, we are proud to say that SheAttack is not far behind the optimal we can possibly achieve.

Table 6: Performance of white-box attacks given shuffled splits.

| | Cora-ML | | CiteSeer | | PubMed | |
|---|---|---|---|---|---|---|
| Attack | Evasion | Poison | Evasion | Poison | Evasion | Poison |
| She-A-Comb | 73.78±1.45 | **71.46±1.07** | 65.39±1.63 | **60.84±1.85** | 77.30±0.37 | 79.12±0.29 |
| PRBCD-shuffle | **71.26±1.29** | 71.89±0.79 | 65.15±2.93 | 63.03±1.87 | 66.48±0.53 | **70.59±0.73** |
| GRBCD-shuffle | 73.73±1.67 | 73.87±1.28 | **64.98±2.69** | 62.67±2.18 | **66.47±0.79** | 71.59±0.70 |
| | Chameleon | | Squirrel | | Roman-Empire | |
| She-A-Comb | **53.11±1.17** | 60.48±0.79 | 34.85±1.25 | 41.25±2.65 | 45.47±0.78 | 48.32±0.75 |
| PRBCD-shuffle | 56.14±2.85 | **56.84±2.36** | 35.06±1.80 | **36.10±1.18** | 43.64±0.71 | **44.41±0.57** |
| GRBCD-shuffle | 56.10±1.91 | 58.90±2.32 | **34.43±0.81** | 38.44±1.17 | **42.24±0.57** | 44.94±0.54 |

**Clustering matters.** Revisiting SheAttack, a key bottleneck of its performance is the clustering process. In our implementation, we adopt Kmeans for its simplicity and efficiency. However, Kmeans requires an additional hyperparameter $k$, which could be tricky to set. And there is still room for improvement as Kmeans could be suboptimal in node clustering (Tsitsulin et al., 2020; Bo et al., 2020). To see the relationship between node clustering and the performance of SheAttack, we add two baselines She-Soft and She-White. She-Soft uses node embeddings generated by supervised GCN as input to generate clusters. She-White directly uses ground-truth labels as clusters. The results are given in the Appendix. On the three citation datasets, especially PubMed, the performance of SheAttack is heavily related to the quality of clusters. This observation motivates us to use advanced node clustering algorithms for improvement, which we leave to future work. We provide the robustness analysis of our method by offering a sensitivity analysis of $k$ in the Appendix F.3.

# 8 CONCLUSION

We study non-targeted graph structural attacks under the RBA setting. To conduct general and effective attacks, we introduce Silhouette Score to measure the difficulty of node classification on graphs. We proposed SheAttack based on a modified Silhouette Score that better fits the attack scenario. We theoretically analyze the change in distances between nodes, verifying the generalizability of SheAttack. We examine the effectiveness of SheAttack on synthetic and real-world datasets with different homophily levels. Our results verify that SheAttack is powerful when compiled with a proper clustering algorithm, and the gap between SheAttack and white-box attacks is narrow when the knowledge of attacks is limited.

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

## A CODE FOR REPRODUCIBILITY

We provide code in the following link: `https://anonymous.4open.science/r/SheAttack-App--85D4/`. Code excluding datasets is provided in the supplementary material.

## B PESUDOCODE OF SHEATTACK

---
**Algorithm 1:** SheAttack

---
**Input:** Graph $(\mathbf{X}, \mathbf{A})$, Loss function $\mathcal{L}_{\text{atk}}$, Budget $B$
**Parameter:** Project dimension $p$, Num of clusters $k$, Block size $b$, Num of Epochs $E$
**Output:** Perturbed adjacency matrix $\mathbf{A}'$

---
1 Compute projected features $\mathbf{X}_{\text{proj}} = \text{PCA}(\mathbf{X})$
2 Compute propagated features $\tilde{\mathbf{X}} = \tilde{\mathbf{A}}^2 \mathbf{X}_{\text{proj}}$
3 Obtain Clusters $\hat{C} = \left\{ \hat{C}_0, \cdots, \hat{C}_{k-1} \right\} = \text{Kmeans}(\tilde{\mathbf{X}})$
4 Initialize $\mathbf{A}' \leftarrow \mathbf{A}$
5 Draw w/o replacement $i_0 \in \{0, 1, \cdots, n^2 - 1\}^b$          // Random block index
6 Set $\Delta_t = B/E$                          // Number of flips per epoch
7 **for** $t \leftarrow 1$ to $E$ **do**
   /* $\mathbf{A} \odot i_0$ means flipping edges in $\mathbf{A}$ whose flattened index is in $i_0$ */
8     Flip top $\Delta_t$ edges with largest $\nabla_{i_{t-1}} \mathcal{L}_{\text{atk}}(\mathbf{X}, \mathbf{A}' \odot i_{t-1}, \hat{C})$
9     Resample $i_t \in \{0, 1, \cdots, n^2 - 1\}^b$
10 **end**
11 **return** $\mathbf{A}'$

---

## C MORE ABOUT EXPERIMENT SETTINGS

Our implementation is based on Pytorch_Geomertric Fey & Lenssen (2019) and GreatX (`https://github.com/EdisonLeeeee/GreatX`).

On small graphs, we implement Kmeans based on scikit-learn `https://scikit-learn.org/stable/modules/generated/sklearn.cluster.KMeans.html#sklearn.cluster.KMeans`. For Kmeans on large graphs, we use kmeans_pytorch with cosine similarity as distance metric for acceleration (`https://github.com/subhadarship/kmeans_pytorch`).

### C.1 DEVICE INFORMATION

All our experiments are conducted on a machine with an NVIDIA A100-SXM4 (80GB memory), Intel Xeon CPU (2.30 GHz), and 512GB of RAM.

### C.2 DATATSET STATISTICS

Dataset information is summarized in Table 7. The homophily metric is the edge-homophily defined in Zhu et al. (2020). The edges are reported on undirected versions, with each undirected edge counted twice. For cSBM datasets, we use a dense split that train/val/test is 60%/20%/20% following (Chien et al., 2021). For homophilic datasets, we generate splits following (Zügner et al., 2018) with train/val/test being 10%/10%/80%. For heterophilic datasets, we choose train/val/test to be 60%/20%/20% for Chameleon and Squirrel following (Chien et al., 2021) and 50%/25%/25% for Roman-Empire following (Platonov et al., 2023).

Table 7: Dataset statistics.

| Dataset | Cora-ML | CiteSeer | PubMed | Chameleon | Squirrel | Roman-Empire | Ogbn-arxiv | Ogbn-products |
|---|---|---|---|---|---|---|---|---|
| Nodes | 2,810 | 2,110 | 19,717 | 2,277 | 5,201 | 22,662 | 169,343 | 2,449,029 |
| Edges | 15,962 | 7,336 | 88,648 | 62,792 | 396,846 | 65,864 | 1,166,243 | 123,718,280 |
| Features | 2,879 | 3,703 | 500 | 2,325 | 2,089 | 300 | 128 | 100 |
| Classes | 7 | 6 | 3 | 5 | 5 | 18 | 40 | 47 |
| Homophily | 0.78 | 0.74 | 0.80 | 0.23 | 0.22 | 0.05 | 0.66 | 0.81 |

## C.3 HYPERPARAMETERS & TRAINING DETAILS

In this section, we report the hyperparameters of attacks and defense models.

**General Settings.** For methods that adopt Greedy Randomized Block Coordinate Descent for optimization, we project raw features into $p = 32$ dimensions. On ogbn-products, the number is set to 16 to avoid out-of-memory issues. The block size is set to be $250,000$ for graphs with less than $10,000$ nodes, $500,000$ for graphs whose number of nodes is between $10,000$ and $100,000$. For large graphs, the block size is set to be $1,000,000$ in ogbn-arxiv and $2,000,000$ in ogbn-products. The edges are greedily flipped over 125 epochs. For methods that rely on clustering, the number of $k$ is set to the ground truth number of classes. (We provide a discussion of the choice of $k$ in F.3.)

For the **victim model**, GCN, we carefully follow the empirical results in the mentioned datasets and tune GCN to have better performance on clean graphs. In all datasets, we set $0.5$ the dropout rate and use Adam optimizer Kingma & Ba (2015). We set the number of convolutional layers to be 2 on small graphs and 3 on large OGB graphs. In Cora-ML, CiteSeer, and PubMed, we set learning rate $0.01$, weight decay $5e - 4$, and hidden size $16$. In Chameleon, Squirrel, and Roman-Empire, we set learning rate $0.05$, weight decay $0.0$, and hidden size $64$. In Ogbn-arxiv and Ogbn-products, we set learning rate $0.01$, weight decay $0.0$, and hidden size $256$. For other defense models, the number of heads of GAT is set to $8$, and other hyperparameters are set the same as GCN.

**Random** and **DICE**: We use the implementation in GreatX for random attacks. These two attacks as random methods include no hyperparameters by default and are generally set to add and delete edges with equal probability. As suggested by Zügner & Günnemann (2019); Li et al. (2023), white-box attacks show a strong tendency to add edges rather than delete. So we include a hyperparameter $\theta$ as the threshold to control the proportion of add/delete in modifications, which is empirically shown useful in improving the performance of random methods. The threshold is chosen from $\{0.5, 1.0\}$, corresponding to the default setting and insertions only. In the experiments, we find that insertions outperform delete in most datasets.

**SPAC** and **SPAC-A**: We implement these two methods following the original paper Lin et al. (2022). We optimize the attack objective for 100 iterations and set adaptive step size for as $lr = \eta * T/\sqrt{t}$, where $\eta$ is the base learning rate, $T$ is the number of epochs and $t$ is the number of the current epoch. For SPAC-A, we choose $128$ lowest eigenvalues and $64$ highest eigenvalues.

**GFAttack**: GFAttack Chang et al. (2020) originates as a targeted attack. We transfer it into an untargeted attack following Lin et al. (2022). We select the top $128$ lowest eigenvalues following the original paper and optimize it similarly to SPAC.

**SelfAttack**: SelfAttack uses a two-layer GCN with a hidden size $16$ as a surrogate model. We use Adam Kingma & Ba (2015) as the optimizer with the learning rate being $0.01$ and weight decay being $5e - 4$ in all datasets. The dropout rate is set to be $0.5$. We train the surrogate model on clusters defined by Kmeans similar to SheAttack (details about clustering are discussed later.). We train the surrogate model on the full graph with 200 epochs and select the best model with the lowest training loss. We use Greedy Randomized Block Coordinate Descent for optimization.

**PEEGA**: PEEGA Li et al. (2022) originally modifies both features and the graph structure with a $O(n^2)$ time and space complexity in optimization. To ensure fairness, we use Greedy Randomized Block Coordinate Descent for optimization and only allow structure perturbations. The vector norm in Shift Loss is set to 2 following the original paper. The regularization term $\lambda$ for Shift loss is tuned between $\{0.0, 1.0, 5.0\}$, where $\lambda = 0.0$ corresponds to the PEEGA version, which only includes homophily loss.

**SheAttack**: For SheAttack, we tune $\lambda$ in the same way in PEEGA. Euclidean distance is applied to measure the distance of embeddings. The clustering is conducted using Kmeans given propagated features on real-world datasets and given raw features on cSBM datasets. Other hyperparameters are specified in the **General Settings** above.

# D   PROOF

## D.1   PROOF OF THEOREM 4.2

To proof Theorem 4.2, we first introduce the following lemma:

**Lemma D.1.** *Given random vectors* $\mathbf{x} \sim N(\boldsymbol{\mu_x}, \boldsymbol{\Sigma_x})$ *and* $\mathbf{y} \sim N(\boldsymbol{\mu_y}, \boldsymbol{\Sigma_y})$*, denote their covariance as* $\boldsymbol{\Sigma_{xy}}$*, the distance between* $\mathbf{x}$ *and* $\mathbf{y}$ *satisfies:*

$$\mathbb{E}\left[\|\mathbf{x}-\mathbf{y}\|^2\right] = \|\boldsymbol{\mu_x}-\boldsymbol{\mu_y}\|^2 + \mathrm{Tr}(\boldsymbol{\Sigma_x}) + \mathrm{Tr}(\boldsymbol{\Sigma_y}) - 2\,\mathrm{Tr}(\boldsymbol{\Sigma_{xy}}).$$

*Proof.*

$$\begin{aligned}
&\|\mathbf{x}-\mathbf{y}\|^2\\
&= \|(\mathbf{x}-\boldsymbol{\mu_x})-(\mathbf{y}-\boldsymbol{\mu_y})+\boldsymbol{\mu_x}-\boldsymbol{\mu_y}\|^2\\
&= \|(\mathbf{x}-\boldsymbol{\mu_x})-(\mathbf{y}-\boldsymbol{\mu_y})\|^2 + 2\left\langle(\mathbf{x}-\boldsymbol{\mu_x})-(\mathbf{y}-\boldsymbol{\mu_y}),\boldsymbol{\mu_x}-\boldsymbol{\mu_y}\right\rangle + \|\boldsymbol{\mu_x}-\boldsymbol{\mu_y}\|^2
\end{aligned}$$

Take the expectations of both sides. Note that the expectation of the second term of the right side is 0. Let $\mathbf{x}' = \mathbf{x} - \boldsymbol{\mu_x}$ and $\mathbf{y}' = \mathbf{y} - \boldsymbol{\mu_y}$, we have

$$\begin{aligned}
\mathbb{E}\left[\|\mathbf{x}-\mathbf{y}\|^2\right] &= \|\boldsymbol{\mu_x}-\boldsymbol{\mu_y}\|^2 + \mathbb{E}\left[\|\mathbf{x}'-\mathbf{y}'\|^2\right]\\
&= \|\boldsymbol{\mu_x}-\boldsymbol{\mu_y}\|^2 + \mathbb{E}\left[\mathbf{x}'^\top\mathbf{x}\right] + \mathbb{E}\left[\mathbf{y}'^\top\mathbf{y}'\right] - 2\mathbb{E}\left[\mathbf{x}'^\top\mathbf{y}'\right]\\
&= \|\boldsymbol{\mu_x}-\boldsymbol{\mu_y}\|^2 + \mathrm{Tr}(\boldsymbol{\Sigma_x}) + \mathrm{Tr}(\boldsymbol{\Sigma_y}) - 2\,\mathrm{Tr}(\boldsymbol{\Sigma_{xy}})
\end{aligned}$$

$\square$

If $\mathbf{y}$ is a constant vector, Lemma D.1 can be written as

$$\mathbb{E}\left[\|\mathbf{x}-\mathbf{y}\|^2\right] = \|\boldsymbol{\mu_x}-\mathbf{y}\|^2 + \mathrm{Tr}(\boldsymbol{\Sigma_x})$$

. Now, we can obtain Theorem 4.2 via direct calculation:

*Proof.* After propagation, the embedding $h_i$ in expectation is:

$$\begin{aligned}
\mathbb{E}[\mathbf{h}_i] &= \frac{pd\boldsymbol{\mu}_1 + qd\boldsymbol{\mu}_2}{pd + qd}\\
&= \frac{p\boldsymbol{\mu}_1 + q\boldsymbol{\mu}_2}{p + q}.
\end{aligned}$$

After perturbation, the expectation of embedding $\mathbf{h}'_i$ becomes:

$$\mathbb{E}[\mathbf{h}'_i] = \frac{(pd + \Delta_1)\boldsymbol{\mu}_1 + (qd + \Delta_2)\boldsymbol{\mu}_2}{pd + qd + \Delta_1 + \Delta_2}.$$

Let $a_i = \|\mathbf{h}'_i - \boldsymbol{\mu}'_1\|^2$, $b_i = \|\mathbf{h}'_i - \boldsymbol{\mu}'_2\|^2$, $c = \mathrm{Tr}\left(\mathrm{Cov}\left(\mathbf{h}'_i\right)\right) = \frac{d}{\sqrt{d_i + \Delta_1 + \Delta_2}}$. By Lemma D.1, we have:

$$\begin{aligned}
\mathbb{E}\left[a_i\right] &= \|\mathbf{h}'_i - \boldsymbol{\mu}'_1\|^2 + \mathrm{Cov}(\mathbf{h}'_i)\\
&= \left\|\frac{(pd + \Delta_1)\boldsymbol{\mu}_1 + (qd + \Delta_2)\boldsymbol{\mu}_2}{pd + qd + \Delta_1 + \Delta_2} - \frac{p\boldsymbol{\mu}_1 + q\boldsymbol{\mu}_2}{p + q}\right\|^2 + c\\
&= \left(\frac{\Delta_1 q - \Delta_2 p}{(p + q)(pd + qd + \Delta_1 + \Delta_2)}\right)^2 \|\boldsymbol{\mu}_1 - \boldsymbol{\mu}_2\|^2 + c.
\end{aligned}$$

$$\mathbb{E}\left[b_i\right] = \|\mathbf{h}_i' - \boldsymbol{\mu}_2'\|^2 + \text{Cov}(\mathbf{h}_i')$$

$$= \left(\frac{p(pd + \Delta_1) - q(qd + \Delta_2)}{(p + q)\left(pd + qd + \Delta_1 + \Delta_2\right)}\right)^2 \|\boldsymbol{\mu}_1 - \boldsymbol{\mu}_2\|^2 + c$$

$\square$

## D.2 Extension to Multi-class Classification

In the main body of the paper, we focus on perturbations in the 2-class classification task. In this section, we offer a discussion about intra-class distance and the preferences of attackers in multi-class classification.

Suppose the number of classes is $c$, and the attacked node $i$ we focus on belongs to class 1 with centroid $\mu_1$. We denote the inter-class probabilities as $q$ for all inter-class pairs following Zhu et al. (2020); Ma et al. (2022b). The centroids and corresponding perturbations are $\boldsymbol{\mu}_2, \boldsymbol{\mu}_3, \cdots, \boldsymbol{\mu}_c$ and $\Delta_2, \Delta_3, \cdots, \Delta_c$. For intra-class probability and corresponding perturbations, we adopt $p$ and $\Delta_1$ as notations. The average node degree is $d$ as in the 2-class case.

After propagation, the embedding $h_i$ in expectation is:

$$\mathbb{E}[\mathbf{h}_i] = \frac{pd\boldsymbol{\mu}_1 + \sum_{j=2}^c qd\boldsymbol{\mu}_j}{pd + (c-1)q}$$

$$= \frac{p\boldsymbol{\mu}_1 + q\sum_{j=2}^c \boldsymbol{\mu}_j}{p + (c-1)q}.$$

The propagated cluster centroid $\boldsymbol{\mu}_0'$ becomes:

$$\boldsymbol{\mu}_1' = \frac{p\boldsymbol{\mu}_1 + q\sum_{j=2}^c \boldsymbol{\mu}_j}{p + (c-1)q}.$$

For class $t \neq 1$, its propagated centroid $\mu_t'$ is:

$$\boldsymbol{\mu}_t' = \frac{p\boldsymbol{\mu}_t + q\sum_{j=2}^c \boldsymbol{\mu}_j}{p + (c-1)q}.$$

After perturbation, the expectation of embedding $\mathbf{h}'$ becomes:

$$\mathbb{E}[\mathbf{h}_i'] = \frac{(pd + \Delta_1)\boldsymbol{\mu}_1 + \sum_{j=2}^c (qd + \Delta_j)\boldsymbol{\mu}_j}{pd + (c-1)qd + \sum_{j=1}^c \Delta_j}.$$

Let $cns = \frac{d}{\sqrt{d_i + \sum_{i=1}^t \Delta_i}}$, the intra-class distance after perturbations is:

$$\mathbb{E}\left[\|\mathbf{h}_i' - \boldsymbol{\mu}_1'\|^2\right] = \left(\frac{(pd + \Delta_1)\boldsymbol{\mu}_1 + \sum_{j=2}^c (qd + \Delta_j)\boldsymbol{\mu}_j}{pd + (c-1)qd + \sum_{j=1}^c \Delta_j} - \frac{p\boldsymbol{\mu}_1 + q\sum_{j=2}^c \boldsymbol{\mu}_j}{p + (c-1)q}\right)^2 + cns$$

$$= \left(\frac{p\left(\sum_{j=2}^c \Delta_j\left(\boldsymbol{\mu}_j - \boldsymbol{\mu}_1\right)\right) + q\left((c-1)\sum_{j=1}^c \Delta_j\boldsymbol{\mu}_j - \sum_{j=1}^c \Delta_j \sum_{j=2}^c \boldsymbol{\mu}_j\right)}{(p + (c-1)q)\left(pd + (c-1)qd + \sum_{j=1}^c \Delta_j\right)}\right)^2 + cns.$$

When $c = 3$, the above equality becomes:

$$\mathbb{E}\left[\|\mathbf{h}_i' - \boldsymbol{\mu}_1'\|\right] = \left(\frac{(p\Delta_2 - q\Delta_1)(\boldsymbol{\mu}_1 - \boldsymbol{\mu}_2) + (p\Delta_3 - q\Delta_1)(\boldsymbol{\mu}_1 - \boldsymbol{\mu}_2) + (q\Delta_2 - q\Delta_3)(\boldsymbol{\mu}_2 - \boldsymbol{\mu}_3)}{(p + 2q)\left(pd + 2qd + \Delta_1 + \Delta_2 + \Delta_3\right)}\right)^2 + cns.$$

From the above equation, we see that removing intra-class edges is still preferred by the attacker to enlarge intra-class distances. Still, low-degree nodes are more fragile. The results for adding edges are more complex as more distance terms between centroids are included. Defining inter-class distance faces the same challenge in dealing with a growing number of cluster centroids. Following previous works addressing heterophily and class-wise distances Ma et al. (2022b); Chen et al. (2022); Platonov et al. (2022), we infer that attacks minimizing inter-class distances seek for perturbations that better disturb the class-wise neighborhood distribution. For example, in a 2-class classification task on heterophilic graphs, the neighborhood distribution of nodes reflects a high preference for inter-class neighbors. As we discussed in the main body of the paper, attacks that shorten inter-class distances on these graphs result in a homophily increase, which can be regarded as a perturbation to nodes' neighborhood distribution. A complete theoretical analysis of this case is left to future work.

# E    MORE ABOUT CSBM DATASETS

We generate cSBM datasets following Deshpande et al. (2018); Chien et al. (2021). Denote a cSBM graph as $\mathcal{G} \sim \text{cSBM}(n, f, \lambda, \mu)$, where $n$ is the number of nodes, $f$ is the dimension of features and $\lambda$ and $\mu$ are hyperparameters controlling the proportion of helpful information between the graph structure and node features.

For a two-class cSBM model with equal class size, denote the node labels as $y_i \in \{-1, +1\}$. Node features are generated following Gaussian distribution $x_i = \sqrt{\frac{\mu}{n}} y_i u + \frac{Z_i}{\sqrt{f}}$, where $u \sim N(0, \mathbf{I}/f)$ and $Z$ is a random noise term. Given an average degree $d$, the graph structure of the cSBM graph is:

$$\mathrm{P}\left[\mathbf{A}_{ij} = 1\right] = \begin{cases} \frac{d + \lambda\sqrt{d}}{n} & \text{if } y_i y_j > 0 \\ \frac{d - \lambda\sqrt{d}}{n} & \text{otherwise.} \end{cases}$$

Let $\Phi = \arctan\left(\frac{\lambda}{\mu}\sqrt{\frac{n}{f}}\right) * \frac{2}{\pi}$. A larger $|\Phi|$ implies a larger $\lambda$ over $\mu$, and the information comes more from the graph structure. A positive $\Phi$ suggests a homophilic graph generated, and a heterophilic graph if $\Phi$ is negative. Note that in this setting, both homophilic and heterophilic graphs have the same amount of useful information as long as the $|\Phi|$ are the same.

In this experiment, we consider a two-class cSBM model with equal class size, where $n = 5000$, $f = 2000$, and $d = 5$ following Chien et al. (2021). We test the attack performance on graphs generated by this model with $\Phi \in \{-0.50, -0.25, 0.25, 0.50\}$ on GCN. The statistics of these models about edge homophily are summarized in the following table: Additional attack results with 0.10 the

Table 8: Statistics of cSBM datasets.

| $\Phi$ | -0.50 | -0.25 | 0.25 | 0.50 |
|---|---|---|---|---|
| Homophily | 0.17 | 0.33 | 0.67 | 0.82 |

perturb ratio are presented in Table 9.

Table 9: Performance of GCN on perturbed cSBM graphs with perturb ratio 0.10. The datasets are named cSBM($\Phi * 100$). The best results are bold, and the second-best results are underlined.

| Attack | cSBM+25 | | cSBM-25 | | cSBM+50 | | cSBM-50 | |
|---|---|---|---|---|---|---|---|---|
| | Evasion | Poisoning | Evasion | Poisoning | Evasion | Poisoning | Evasion | Poisoning |
| Clean | 81.25±0.32 | | 57.42±1.00 | | 93.39±0.18 | | 74.11±0.30 | |
| Random | 80.50±0.91 | 79.73±1.14 | 56.56±0.76 | 57.68±0.28 | 92.03±0.53 | 91.76±0.52 | 72.83±1.10 | 72.00±0.89 |
| DICE | 76.72±0.48 | 75.01±0.73 | 59.74±0.81 | 62.26±0.67 | **88.98±0.61** | **88.08±0.40** | 79.36±0.21 | 80.13±0.32 |
| SPAC-A | 80.90±0.23 | 81.07±0.34 | 57.41±1.13 | 57.52±0.29 | 93.26±0.50 | 93.50±0.22 | 75.50±0.17 | 75.39±0.28 |
| GFAttack | 81.10±0.39 | 80.30±1.03 | 58.00±0.78 | 57.89±0.95 | 92.45±0.49 | 92.21±0.45 | 73.78±0.41 | 74.00±0.75 |
| SelfAttack | 79.12±0.57 | 78.54±0.43 | 57.58±0.64 | 58.66±0.90 | 91.87±0.36 | 91.73±0.47 | 74.90±0.52 | 74.88±0.70 |
| PEEGA | 81.58±0.34 | 81.22±0.59 | 58.06±0.62 | 57.49±0.89 | 93.09±0.43 | 93.20±0.43 | 73.28±0.59 | 72.42±0.48 |
| PEEGA-Comb | 79.10±0.66 | 77.95±0.77 | 57.07±0.75 | 57.04±0.56 | 90.72±0.50 | 89.52±0.67 | 75.44±0.33 | 75.14±0.54 |
| She | 74.05±0.98 | 70.62±0.71 | **51.62±0.83** | 49.90±0.50 | 90.48±1.12 | 89.54±1.35 | **70.86±1.07** | **69.71±0.84** |
| She-Comb | 74.64±0.51 | 70.91±0.52 | 53.50±0.81 | 52.26±0.53 | 90.34±1.07 | 89.42±0.55 | 71.90±0.65 | 70.50±1.12 |
| She-A | **73.60±0.51** | **70.27±0.64** | 51.97±1.12 | **49.79±1.12** | 90.86±1.27 | 90.40±1.30 | 71.31±0.56 | 70.10±0.81 |
| She-A-Comb | 73.82±0.55 | 70.72±1.00 | 52.53±0.81 | 49.90±0.55 | 91.06±0.77 | 90.54±1.07 | 71.41±0.64 | 69.90±0.85 |

# F    ADDITIONAL EXPERIMENT RESULTS

## F.1    ATTACK LARGE OGB DATASETS

We conduct experiments on large datasets ogbn-arxiv and ogbn-products Hu et al. (2020). Notably, ogbn-products includes more than 2,000,000 nodes and more than 60,000,000 edges. The results are provided in Table 10 and Table 11, verifying both the effectiveness and efficiency of SheAttack.

Table 10: Performance of GCN on ogbn-arxiv with perturb ratio 0.10. The experiment is repeated only once on the fixed split.

| Attack | Evasion | Poisoning |
|---|---|---|
| Clean | 69.60 | |
| Random | 66.04 | 65.34 |
| DICE | 64.78 | 64.59 |
| PEEGA | 66.92 | 67.38 |
| PEEGA-Comb | 63.52 | 64.65 |
| She-A | 63.37 | 63.63 |
| She-A-Comb | **61.64** | **62.72** |

Table 11: Performance of GCN on ogbn-products with perturb ratio 0.10. Experiments are repeated once.

| Attack | Evasion | Poisoning |
|---|---|---|
| Clean | 75.80 | |
| Random | 68.67 | 70.29 |
| PEEGA | 70.28 | 72.77 |
| PEEGA-Comb | 63.80 | 67.76 |
| She-A | 63.84 | 67.40 |
| She-A-Comb | **62.93** | **67.28** |

## F.2 PERFORMANCE OF SHEATTACK UNDER DIFFERENT SETTINGS

In Table 12, we provide the performance of SheAttack given different clusters. She-Soft uses node embeddings generated by supervised GCN as input to generate clusters. Specifically, we use the output of the victim two-layer GCN, denoted as $Z \in \mathbb{R}^{n \times c}$, as the input of Kmeans algorithms. Since GCN output embeddings are highly distinguishable, we believe it yields better clustering results than inputting the propagated projected features. She-White directly uses ground-truth labels as clusters, utilizing the same amount of available information as DICE, where only victim models and training/test splits are unknown. From Table 12, we see that She-White leads to surprisingly high performance, even outperforming results in shuffled white-box attacks. If we use advanced clustering methods or apply unsupervised learning methods to obtain better inputs for Kmeans, SheAttack is possible to obtain further performance improvement.

Table 12: Performance of SheAttack under different settings

| Dataset | Cora-ML | | CiteSeer | | PubMed | |
|---|---|---|---|---|---|---|
| | Evasion | Poisoning | Evasion | Poisoning | Evasion | Poisoning |
| She-Black | 74.72±1.46 | 73.20±0.94 | 66.02±1.34 | 62.67±0.94 | 77.79±0.70 | 79.70±0.13 |
| She-Soft | 75.23±1.48 | 72.91±1.66 | 65.75±1.26 | 60.83±1.76 | 73.28±1.02 | 76.93±0.39 |
| She-White | **72.27±1.46** | **70.34±0.92** | **64.56±1.62** | **59.30±1.37** | **69.71±0.82** | **74.92±0.10** |

## F.3 SENSITIVITY ANALYSIS OF $k$ IN KMEANS

In Figure 4, we offer a sensitivity analysis for SheAttack w.r.t the number of clusters $k$ in Kmeans. In all four datasets, the performance of SheAttack is stable as long as $k$ is not too small compared with the number of classes. For example, the performance of SheAttack is stable after $k$ exceeds 4 in Cora-ML. In ogbn-arxiv, SheAttack even benefits when $k$ grows larger than the ground-truth number of classes. In the experiments, we did not detailedly tune the number of $k$ in Kmeans for fairness. But we can determine $k$ by choosing a proper $k$ with a larger Silhouette Score or using other adaptive clustering methods for improvement as discussed in F.2.

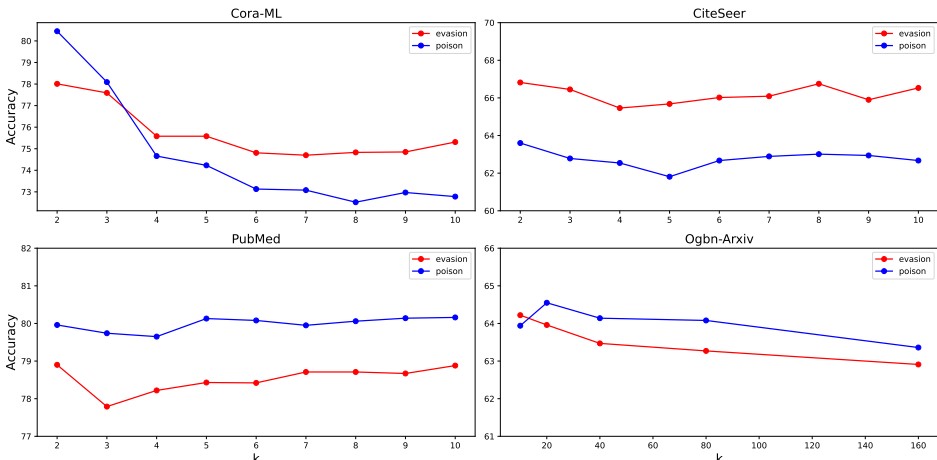

Figure 4: Attack performance with different $k$.

## F.4 TRANSFER TO OTHER DEFENSE MODELS

We transfer the attack results to other defense models, including GAT Velickovic et al. (2018), RobustGCN Zhu et al. (2019), MedianGCN Chen et al. (2021) and GNNGuard Zhang & Zitnik (2020). All experiments are conducted on datasets with a 0.20 perturb ratio. We only report the results on homophilic datasets as these methods do not provide guarantees in defending heterophilic graphs. The authors in GNNGuard suggest using embeddings based on structural roles to handle heterophilic settings, yet being impractical for its complexity and neglecting original node features. In fact, we discover that these methods perform inferior to GCN in clean heterophilic datasets and therefore perform even worse than GCN under attacks. We believe the attacks and defenses under heterophily are still topics that are in progress.

The results are summarized in Table 13, Table 14, Table 15 and Table 16. Generally, SheAttack is effective in lowering the performance of all defense models. As the perturbed graphs are transferred from attacking GCN for white-box attacks, we see that black-box attacks are not far behind white-box attacks. In Cora-ML and CiteSeer, SheAttack even outperforms white-box attacks (not shuffled) when attacking GNNGuard.

Table 13: Performance of GAT on perturbed graphs with perturb ratio 0.20.

| | Cora-ML | | CiteSeer | | PubMed | |
|---|---|---|---|---|---|---|
| Attack | Evasion | Poisoning | Evasion | Poisoning | Evasion | Poisoning |
| Clean | 85.88±0.46 | | 73.59±1.20 | | 85.67±0.27 | |
| Random | 81.11±1.35 | 81.48±0.86 | 69.73±0.95 | 68.79±0.89 | 80.90±0.55 | 81.71±0.33 |
| DICE | 78.62±1.44 | 77.96±1.32 | 67.51±0.92 | 66.37±1.01 | **75.44±0.83** | **76.71±0.20** |
| SPAC | 82.68±0.72 | 81.97±0.39 | 72.49±1.10 | 72.00±1.45 | * | * |
| SPAC-A | 81.67±1.07 | 82.94±0.44 | 71.20±1.30 | 71.49±0.79 | * | * |
| GFAttack | 81.71±1.23 | 81.45±0.67 | 70.77±0.88 | 70.27±2.24 | * | * |
| SelfAttack | 81.25±0.57 | 80.97±0.48 | 73.55±0.93 | 73.92±0.91 | 81.27±0.47 | 81.63±0.39 |
| PEEGA | 82.82±0.79 | 82.64±0.61 | 69.35±0.72 | 70.24±1.15 | 84.68±0.31 | 84.65±0.47 |
| PEEGA-Comb | 75.56±1.27 | 73.89±0.88 | 65.33±1.03 | 62.35±1.36 | 76.74±1.06 | 79.05±0.52 |
| She | 76.41±1.02 | 75.71±0.59 | 65.50±1.26 | 63.68±2.09 | 77.70±1.03 | 79.91±0.40 |
| She-Comb | 74.69±1.07 | 73.07±1.11 | 65.21±1.20 | 62.12±1.03 | 76.40±1.38 | 79.26±0.67 |
| She-A | 76.83±0.99 | 75.58±0.73 | 66.74±0.98 | 64.91±1.42 | 80.06±0.70 | 81.37±0.47 |
| She-A-Comb | **73.91±0.90** | **71.76±0.48** | **65.06±1.27** | **61.61±1.80** | 76.88±0.82 | 79.20±0.79 |
| PRBCD | 64.55±1.35 | 60.93±1.79 | 59.89±2.64 | 56.48±2.70 | 59.38±0.62 | 59.70±0.47 |
| GRBCD | 70.90±1.78 | 67.75±0.91 | 60.81±2.21 | 57.58±1.81 | 60.38±0.79 | 60.20±0.46 |

Table 14: Performance of RobustGCN on perturbed graphs with perturb ratio 0.20.

| Attack | Cora-ML | | CiteSeer | | PubMed | |
|---|---|---|---|---|---|---|
| | Evasion | Poisoning | Evasion | Poisoning | Evasion | Poisoning |
| Clean | 83.59 ± 6.01 | | 74.75 ± 0.58 | | 86.04±0.26 | |
| Random | 79.76±5.60 | 79.63±5.79 | 71.03±0.63 | 70.00±1.83 | 82.66±0.20 | 79.82±5.44 |
| DICE | 77.54±5.91 | 76.20±6.18 | 68.68±1.10 | 67.84±0.72 | 78.07±0.40 | **71.08±5.73** |
| SPAC | 81.01±5.76 | 78.54±5.94 | 73.23±0.57 | 72.26±0.73 | * | * |
| SPAC-A | 80.20±5.59 | 79.71±5.66 | 73.61±0.72 | 73.13±0.96 | * | * |
| GFAttack | 80.27±6.00 | 78.90±7.20 | 71.50±1.34 | 70.89±1.74 | * | * |
| SelfAttack | 78.96±5.68 | 77.72±6.42 | 74.21±0.62 | 74.63±0.60 | 81.73±0.69 | 78.80±4.93 |
| PEEGA | 80.95±5.95 | 79.67±5.38 | 71.50±0.82 | 70.28±1.75 | 85.02±0.32 | 84.74±0.30 |
| PEEGA-Comb | 73.88±6.36 | 70.37±6.20 | 65.05±1.99 | 62.70±0.93 | 78.32±0.67 | 72.53±5.73 |
| She | 73.51±4.81 | 71.62±4.12 | 65.30±1.57 | 63.58±1.58 | 78.27±0.73 | 75.34±5.02 |
| She-Comb | 73.19±4.78 | 70.34±4.46 | 64.62±1.89 | 62.95±1.42 | **77.79±0.77** | 72.70±5.47 |
| She-A | 74.23±5.00 | 71.74±4.62 | 67.56±1.04 | 65.65±1.64 | 80.24±0.19 | 78.33±4.40 |
| She-A-Comb | **72.68±5.15** | **69.88±4.78** | **64.59±2.32** | **61.29±2.03** | 77.86±0.36 | 72.51±5.00 |
| PRBCD | 59.87±4.00 | 57.56±3.67 | 60.28±2.62 | 56.47±2.54 | 62.59±0.71 | 62.24±0.75 |
| GRBCD | 65.55±4.06 | 62.78±5.11 | 59.48±2.81 | 58.41±2.60 | 62.09±0.78 | 62.24±0.99 |

Table 15: Performance of MedianGCN on perturbed graphs with perturb ratio 0.20.

| Attack | Cora-ML | | CiteSeer | | PubMed | |
|---|---|---|---|---|---|---|
| | Evasion | Poisoning | Evasion | Poisoning | Evasion | Poisoning |
| Clean | 85.55 ± 0.39 | | 74.44±1.87 | | 84.97±0.25 | |
| Random | 82.17±0.80 | 81.33±0.71 | 72.52±1.02 | 70.98±1.60 | 82.36±0.37 | 82.02±0.51 |
| DICE | 80.02±0.49 | 78.23±0.70 | 70.77±1.11 | 67.88±0.85 | **77.38±0.33** | **76.76±0.25** |
| SPAC | 82.00±0.69 | 81.10±0.59 | 72.57±1.73 | 70.59±1.58 | * | * |
| SPAC-A | 79.86±0.64 | 80.20±0.43 | 71.64±1.20 | 70.78±1.31 | * | * |
| GFAttack | 82.59±0.47 | 81.75±0.51 | 72.90±1.18 | 73.19±0.68 | * | * |
| SelfAttack | 81.57±0.59 | 80.20±0.84 | 74.18±1.75 | 73.74±1.39 | 80.78±0.32 | 80.77±0.37 |
| PEEGA | 83.73±0.53 | 82.23±0.63 | 71.95±1.31 | 70.49±0.53 | 83.90±0.16 | 84.01±0.29 |
| PEEGA-Comb | 78.83±0.44 | 75.82±1.37 | 70.84±1.70 | 66.53±1.62 | 80.22±0.33 | 79.99±0.41 |
| She | 75.12±0.86 | 73.56±0.90 | 69.11±1.20 | 65.55±0.65 | 78.34±0.24 | 78.39±0.25 |
| She-Comb | 76.09±1.06 | 73.02±1.02 | 70.40±1.59 | 66.65±1.47 | 78.67±0.40 | 78.35±0.20 |
| She-A | 75.59±0.94 | 74.00±1.08 | **68.66±1.13** | 65.60±1.72 | 80.00±0.28 | 79.84±0.20 |
| She-A-Comb | **75.34±0.86** | **71.61±1.87** | 70.36±1.71 | **65.26±1.43** | 78.80±0.35 | 78.76±0.19 |
| PRBCD | 66.23±0.43 | 62.09±1.42 | 61.62±1.66 | 56.71±2.14 | 67.95±0.33 | 67.67±0.44 |
| GRBCD | 68.44±1.29 | 66.95±1.79 | 61.02±1.73 | 58.34±2.22 | 66.56±0.38 | 66.10±0.40 |

Table 16: Performance of GNNGuard on perturbed graphs with perturb ratio 0.20.

| Attack | Cora-ML | | CiteSeer | | PubMed | |
|---|---|---|---|---|---|---|
| | Evasion | Poisoning | Evasion | Poisoning | Evasion | Poisoning |
| Clean | 79.19±1.06 | | 71.75±1.65 | | 86.71±0.39 | |
| Random | 78.84±1.18 | 79.09±1.68 | 71.32±1.55 | 70.82±1.37 | 85.82±0.37 | 85.62±0.32 |
| DICE | 78.24±0.90 | 77.96±1.36 | 70.69±1.39 | 69.61±0.94 | 84.63±0.36 | 84.37±0.33 |
| SPAC | 78.50±0.99 | 78.33±1.79 | 70.57±1.51 | 70.33±0.89 | * | * |
| SPAC-A | 78.67±1.16 | 79.01±1.47 | 71.11±1.74 | 70.66±1.54 | * | * |
| GFAttack | 79.15±0.99 | 79.56±1.28 | 71.60±1.87 | 71.04±1.26 | * | * |
| SelfAttack | 78.60±1.04 | 78.50±1.22 | 71.55±1.76 | 70.92±1.37 | 85.52±0.39 | 85.41±0.30 |
| PEEGA | 79.27±0.97 | 79.50±1.18 | 71.30±1.63 | 71.00±1.46 | 86.08±0.37 | 86.04±0.34 |
| PEEGA-Comb | 79.18±1.06 | 79.40±1.17 | 71.69±1.68 | 71.69±1.63 | 86.31±0.36 | 86.32±0.26 |
| She | 77.31±1.05 | 77.33±1.20 | **69.16±1.69** | 67.86±1.46 | 84.26±0.37 | 84.47±0.22 |
| She-Comb | 77.94±1.18 | 78.04±1.43 | 71.68±1.70 | 71.78±1.18 | 84.87±0.36 | 85.09±0.20 |
| She-A | **76.39±1.38** | **76.34±1.52** | 69.17±1.45 | **67.17±1.46** | **83.38±0.31** | **83.52±0.28** |
| She-A-Comb | 77.60±1.10 | 77.51±1.53 | 71.66±1.69 | 71.48±1.81 | 83.67±0.41 | 83.70±0.17 |
| PRBCD | 78.19±1.07 | 78.65±1.43 | 69.93±1.34 | 70.20±0.77 | 82.58±0.36 | 82.61±0.28 |
| GRBCD | 78.55±1.10 | 79.17±1.39 | 69.40±1.40 | 69.63±1.47 | 82.09±0.33 | 82.14±0.35 |

## F.5 RESULTS UNDER DIFFERENT PERTURB RATES

We report the performance of attacks with perturb rates to be 0.10 and 0.15 in the small datasets. The results are summarized in Table 17, Table 18, Table 19 and Table 20.

Table 17: Performance of GCN on perturbed graphs with perturb ratio 0.10.

| Attack | Cora-ML | | CiteSeer | | PubMed | |
|---|---|---|---|---|---|---|
| | Evasion | Poisoning | Evasion | Poisoning | Evasion | Poisoning |
| Clean | 85.94±0.74 | | 74.09±1.47 | | 86.59±0.26 | |
| Random | 83.85±1.12 | 83.81±1.08 | 72.69±1.32 | 71.16±1.32 | 84.51±0.24 | 84.54±0.29 |
| DICE | 82.78±1.31 | 82.48±1.07 | 71.61±1.32 | 70.17±1.12 | 81.66±0.42 | 81.92±0.19 |
| SPAC | 84.55±0.80 | 84.26±0.60 | 73.00±1.34 | 72.76±1.10 | * | * |
| SPAC-A | 84.40±0.83 | 84.07±0.89 | 73.34±1.41 | 73.21±1.45 | * | * |
| GFAttack | 83.76±0.97 | 84.25±0.76 | 73.27±1.49 | 72.63±1.46 | * | * |
| SelfAttack | 83.40±0.85 | 82.72±0.62 | 73.87±1.41 | 73.83±2.11 | 83.69±0.40 | 83.82±0.09 |
| PEEGA | 84.92±0.76 | 84.84±0.65 | 73.05±1.63 | 72.89±1.55 | 86.00±0.30 | 85.89±0.24 |
| PEEGA-Comb | 80.18±0.87 | 79.21±1.24 | **69.04±1.45** | 66.90±1.60 | 81.23±0.44 | 81.99±0.20 |
| She | 79.23±1.19 | 78.06±1.52 | 69.79±1.25 | 66.88±1.36 | 81.52±0.49 | 82.42±0.23 |
| She-Comb | 79.14±1.17 | 77.46±0.96 | 69.06±1.61 | **66.45±1.80** | **80.57±0.46** | **81.83±0.39** |
| She-A | 80.12±1.17 | 78.78±1.00 | 70.44±1.26 | 68.83±1.66 | 82.89±0.28 | 83.21±0.12 |
| She-A-Comb | **78.99±1.22** | **76.99±0.97** | 69.21±1.65 | 66.46±0.99 | 81.11±0.28 | 81.92±0.26 |

## F.6 RUNNING TIME

In the main body of the paper, we present the complexity analysis of SheAttack. We also offer experiment results on large graphs ogbn-arxiv and ogbn-products, verifying the scalability of SheAttack. In ogbn-products, SheAttack with the approximate version finishes in 586.74(s), with the version including shift loss finishing in 750.84(s). In Table 21, we provide the running time of SheAttack on cSBM datasets. We see that attacks are much faster if eigendecomposition is not required. Even on graphs with 5,000 nodes, spectral methods take around 100 seconds to finish, which is about 20x times slower than other baselines. SheAttack conducts clustering only once and is, therefore, more efficient.

Table 18: Performance of GCN on perturbed graphs with perturb ratio 0.15.

| Attack | Cora-ML | | CiteSeer | | PubMed | |
| | Evasion | Poisoning | Evasion | Poisoning | Evasion | Poisoning |
|---|---|---|---|---|---|---|
| Clean | 85.94±0.74 | | 74.09±1.47 | | 86.59±0.26 | |
| Random | 83.03±1.12 | 83.01±1.10 | 71.94±1.27 | 70.59±1.15 | 83.50±0.30 | 83.62±0.30 |
| DICE | 81.18±1.26 | 80.88±0.54 | 70.55±1.33 | 68.70±1.24 | 79.54±0.32 | **80.02±0.08** |
| SPAC | 83.64±0.79 | 83.20±0.70 | 72.62±1.20 | 71.45±1.32 | * | * |
| SPAC-A | 83.63±1.42 | 84.10±1.11 | 73.14±1.48 | 72.68±0.98 | * | * |
| GFAttack | 83.36±1.20 | 83.46±0.82 | 72.83±1.67 | 71.95±1.12 | * | * |
| SelfAttack | 82.52±0.98 | 81.37±0.50 | 73.60±1.57 | 73.25±1.89 | 82.74±0.46 | 83.05±0.10 |
| PEEGA | 84.09±0.98 | 83.91±1.14 | 72.27±1.46 | 71.23±0.96 | 85.72±0.30 | 85.61±0.33 |
| PEEGA-Comb | 78.33±1.14 | 76.73±1.46 | 67.73±1.37 | 64.47±1.21 | 79.33±0.87 | 80.92±0.16 |
| She | 76.98±1.54 | 75.24±1.48 | 68.05±1.25 | 65.41±0.70 | 79.45±0.62 | 80.98±0.25 |
| She-Comb | 76.73±1.53 | 74.72±1.60 | 67.68±1.48 | **63.73±0.30** | **78.76±0.56** | 80.62±0.15 |
| She-A | 77.94±1.67 | 76.17±1.16 | 68.66±0.99 | 66.72±0.48 | 81.41±0.37 | 81.77±0.41 |
| She-A-Comb | **76.64±1.50** | **73.74±1.45** | **67.35±1.51** | 63.93±1.80 | 79.11±0.38 | 80.22±0.31 |

Table 19: Performance of GCN on heterophilic graphs with perturb ratio 0.10.

| Attack | Chameleon | | Squirrel | | Roman-Empire | |
| | Evasion | Poisoning | Evasion | Poisoning | Evasion | Poisoning |
|---|---|---|---|---|---|---|
| Clean | 67.59±2.48 | | 53.22±1.80 | | 49.85±0.54 | |
| Random | 64.17±2.10 | 61.58±2.36 | 43.11±2.70 | **41.02±2.03** | 49.63±0.66 | 50.11±0.56 |
| DICE | 62.37±1.69 | **60.13±1.70** | 43.19±1.86 | 43.23±1.00 | 49.33±0.57 | 49.56±0.37 |
| SPAC | 62.46±0.98 | 61.27±2.16 | 42.75±1.46 | 41.08±1.63 | * | * |
| SPAC-A | 64.78±0.63 | 63.90±1.88 | 45.28±1.91 | 45.57±1.23 | * | * |
| GFAttack | 64.12±1.52 | 62.15±1.85 | 44.65±1.65 | 44.36±1.76 | * | * |
| SelfAttack | 63.11±1.47 | 64.82±1.80 | 44.78±0.66 | 46.74±1.61 | 48.50±0.55 | 48.77±0.29 |
| PEEGA | 62.50±1.74 | 66.10±2.66 | 48.86±2.66 | 50.30±1.86 | 50.57±0.44 | 51.84±0.66 |
| PEEGA-Comb | 61.71±1.57 | 62.85±1.25 | 41.59±1.79 | 45.42±2.23 | 47.48±0.63 | 49.21±0.31 |
| She | 60.18±2.07 | 62.06±2.32 | 44.07±2.01 | 43.84±1.89 | **46.78±0.64** | 48.27±0.45 |
| She-Comb | 60.39±0.84 | 62.59±2.49 | 43.80±1.56 | 43.69±1.28 | 47.32±0.33 | 48.58±0.60 |
| She-A | **58.73±1.83** | 61.84±3.36 | 41.23±1.09 | 44.59±2.21 | 47.29±0.37 | 48.33±0.40 |
| She-A-Comb | 59.47±1.27 | 63.55±1.43 | **40.50±2.15** | 45.21±1.39 | 47.30±0.55 | **48.22±0.43** |

Table 20: Performance of GCN on heterophilic graphs with perturb ratio 0.15.

| Attack | Chameleon | | Squirrel | | Roman-Empire | |
| | Evasion | Poisoning | Evasion | Poisoning | Evasion | Poisoning |
|---|---|---|---|---|---|---|
| Clean | 67.59±2.48 | | 53.22±1.80 | | 49.85±0.54 | |
| Random | 62.02±2.31 | 59.30±1.62 | 41.04±2.24 | **37.68±1.70** | 49.70±0.72 | 50.44±0.53 |
| DICE | 60.92±2.14 | 60.26±1.02 | 41.25±1.43 | 42.73±1.50 | 49.11±0.75 | 50.22±0.20 |
| SPAC | 59.96±0.95 | **58.68±1.63** | 40.86±1.75 | 39.10±1.99 | * | * |
| SPAC-A | 62.72±1.03 | 63.86±1.70 | 40.50±1.62 | 43.90±0.76 | * | * |
| GFAttack | 63.07±1.31 | 60.57±2.61 | 42.92±1.39 | 43.69±1.74 | * | * |
| SelfAttack | 61.84±1.07 | 62.02±1.26 | 41.25±0.86 | 44.50±0.66 | 48.17±0.33 | 49.10±0.64 |
| PEEGA | 60.18±2.05 | 65.09±2.87 | 46.59±2.30 | 50.72±1.68 | 50.57±0.43 | 53.11±0.61 |
| PEEGA-Comb | 60.22±2.08 | 60.44±1.64 | 38.73±1.79 | 42.59±1.44 | 46.39±0.63 | 49.09±0.26 |
| She | 59.12±2.00 | 61.67±3.02 | 42.27±1.51 | 41.38±2.61 | **45.97±0.29** | **47.61±0.24** |
| She-Comb | 57.68±1.01 | 61.10±1.85 | 41.11±1.68 | 42.48±1.93 | 46.21±0.49 | 48.65±0.78 |
| She-A | **56.10±1.13** | 61.45±1.58 | 37.87±1.76 | 42.86±2.41 | 46.42±0.57 | 47.86±0.47 |
| She-A-Comb | 56.18±0.80 | 62.15±2.05 | **37.68±1.75** | 42.94±2.48 | 46.28±0.41 | 48.46±0.60 |

Table 21: Time cost of attacks over 5 repeated experiments.

| Attack | Time (s) |
|---|---|
| Random | 0.24±0.00 |
| DICE | 0.95±0.04 |
| SelfAttack | 2.91±0.18 |
| PEEGA | 1.99±0.11 |
| PEEGA-Comb | 3.12±0.11 |
| SPAC-A | 89.51±17.51 |
| GFAttack | 109.90±2.01 |
| SheAttack | 2.82±0.18 |
| SheAttack-Comb | 4.06±0.21 |
| SheAttack-A | 2.61±0.16 |
| SheAttack-A-Comb | 3.71±0.18 |

## G  UNNOTICABILITY

Unnoticability is important for an attack to be effective, but the definition is quite vague currently. Here, we calculate unnoticability from four perspectives: change of degree, homophily, spectral difference, and node focus.

The degree change is the first aspect to be studied in terms of unnoticeability in graph structural attacks, and a homophily change is also observed in most attacks (Zügner & Günnemann, 2019; Zhu et al., 2022). Denote the node degree of node $i$ before and after the attack as $d_i$ and $d_i'$. We measure degree change using the relative degree shift which is $\frac{1}{n}\sum_{v_i\in V}\frac{d_i'-d_i}{d_i}$. For homophily, we use edge homophily as the metric to measure the homophily change before and after the attack. The spectral difference has been a major part of spectral methods to work. For spectral change, we calculate the spectral difference following (Lin et al., 2022). And (Li et al., 2023) show that white-box attacks focus on specific parts of nodes which is highly related to the training/test splits.

We summarize the statistics on the Cora-ML dataset with 0.20 the perturb ratio. We choose random attacks that only add edges as the baseline, as white-box attacks and SheAttack all show a tendency to add edges rather than delete in practice. The node focus calculates the proportion of modifications that happen among different node sets. The results are summarized in 22.

Table 22: Comparision about Unnoticability for different attacks. The homophily level after the attack is presented. Node focus is presented as (Train-Train / Train-Test / Test-Test).

| Attack | Degree | Homophily | Spectral | Focus |
|---|---|---|---|---|
| Random (Only-add) | 0.42 | 0.683 | 2.56 | 0.01 / 0.17 / 0.63 |
| SPAC | -0.19 | 0.745 | 7.43 | 0.01 / 0.16 / 0.63 |
| GRBCD (White) | 0.63 | 0.654 | 2.50 | 0.00 / 0.09 / 0.85 |
| SheAttack | 0.61 | 0.661 | 2.19 | 0.01 / 0.16 / 0.63 |
| SheAttack-Comb | 0.59 | 0.657 | 2.33 | 0.01 / 0.17 / 0.64 |

We see that SheAttack and white-box prefer low-degree nodes, and this could result in an increase in the average degree. The homophily level is generally lowered as Cora-ML is homophilic. Spectral methods introduce larger spectral differences, while other methods do not. White-box attacks have a strong tendency to focus on modification related to node splits, while RBAs do not. To conclude, the unnoticability of SheAttack is in an acceptable range.

