# OpenReview forum: "SheAttack: A Silhouette Score Motivated Restricted Black-Box Attack on Graphs"
_ICLR.cc/2024/Conference — Submitted to ICLR 2024_

### Official Review · Reviewer_1uf8 · 2023-10-30

**Soundness:** 2 fair
**Presentation:** 3 good
**Contribution:** 2 fair
**Rating:** 3
**Confidence:** 4

**Summary:**

The paper introduces a new method, SheAttack, for attacking Graph Neural Networks (GNNs) in the Restrict Black-box attack setting. This method aims to diminish the quality of graph data by manipulating node distinguishability. The approach utilizes a Modified Silhouette Score (MSS) to assess graph quality across various homophily levels. Experiments show that SheAttack performs effectively on both homophilic and heterophilic graphs and offers comparable results to more knowledge-intensive white-box attacks.

**Strengths:**

**Clarity**: The paper is well written and clear to understand.

**Quality**: The assumptions made about real-world scenarios and RBA are well-considered.

**Significance**: The problem addressed is significant due to the growing need to detect vulnerabilities in graph neural network models. The problem setting in this paper (RBA) seems more aligned with real-world scenarios compared to white-box and grey-box attacks.

**Weaknesses:**

* The attack proposed relies heavily on node features to achieve a quality cluster to replace ground-truth labels when calculating the Silhouette score. This dependence is a significant vulnerability. If one adds noise to features and incorporates a de-noising mechanism within the base model, the attack's efficacy could be undermined since the attacker wouldn't know about the noise or how to de-noise the features.

* Given that the attacker has access to node features, it might be more impactful to target both the structure and features. Not leveraging this information seems like a missed opportunity.

* The attack lacks a theoretical foundation; it's mainly empirical. There are no guarantees about the efficacy of the attack.

* The paper seems to have limited novelty. The idea of using clustering due to the absence of label information in RBA and the shift loss has been previously explored. The primary innovation appears to be the modification of the Silhouette score, which has its challenges.

* In Section 3.2, the authors propose modifications in $a$ and $b$ to accommodate the absence of ground-truth labels. However, the later modifications in $b$ do not address the issue of pushing nodes of different classes further apart.

* The parameter $\Delta$ plays a significant role in the problem definition (Section 2), yet it isn't discussed in the methodology or experiment sections. It's unclear how much perturbation is excessive or how this was determined and verified.

* Please ensure notation consistency. In Section 2, both notations $f_\theta(X;A)$ and $f_\theta(X,A)$ are used.

* The related work section mentions interesting algorithms not used as benchmarks. Specifically, the absence of some RL-based methods was noticeable. Why were they excluded? Modifications to the benchmark could also be applied to other methods.

**Questions:**

See above.

---

> ### Author Response · Authors · 2023-11-12
> **Reply to Reviewer 1uf8**
>
> Thanks for your time and efforts!
>
> **Reply to the Weaknesses**
> 1. The quality of clusters would be a concern in practice. However, stand at the attacker side, the quality of features would lead to a somewhat chicken and egg paradox.
> If the cluster performance is extremely undesirable, or there exists huge feature noise, the defenders will not be able to achieve desirable classification performance, and therefore no need for attacks.
> Note that the defender have no access to inject controllable noise in most cases. If so, the results of all current attack methods would be unreliable.
> In our empirical results, Kmeans achieves acceptable clustering and lead to successful attacks in most graphs.
> As we discussed in Section 7, improving the quality of clusters is feasible.
> One could adopt Graph Constrative Learning, advanced or dataset-tailored clustering methods to achieve better cluster performance.  And as we discussed in Appendix F.2, such modifications would lead to even better performance for SheAttack.
> 2. Attacking node features would be an opportunity for further improvement.
> It can be easily incorporated into our framework by updating $X$ and $A$ simultaneously through the gradients.
> We do not focus on feature attacks for more fair evaluation as most RBAs only focus on structural attacks.
> 3. It is hard to derive general theoretical results for an RBA to surpass the others in all aspects.
> In our paper, we pay attention to heterophily, which has been ignored in previous RBA design.
> We theoretically verify that SheAttack is suitable for both homophilic and heterophilic settings, which is in accordance with the empirical results in synthetic datasets.
> 4. The novelty of our methods is that we get rid of the homophily assumption and costly spectral techniques.
> We try to induce the graph into a more general dilemma, where nodes are of less distinguishablility from the perspective of the class-wise distance.
> The class-wise distance is more fundamental, yet rarely explored in graph adversarial attacks.
> We provide theoretical interpretation for previous losses, and propose a more general attack framework.
> 5. The modification in $b$ is to enlarge the scope of the attacker to fit in practical attack scenario.
> When considering just clustering, Silhouette Score is good enough.
> But during attacks, the Silhouette Score without modifications tend to ignore opportunities.
> The modifications do not aim to push intra-class nodes further apart, but enlarge the feasible sets during edge flipping.
> The benefits are verified empirically in Table 5 in Section 6.
> 6. We include experiments with different $\Delta$ in Appendix F.5, with the perturb ratio being 10%, 15% and 20%.
> SheAttack constantly hold promising performance.
> 7. Thanks for pointing the typos out. We will ensure the consistency of notations in the revised version.
> 8. The RL-based models mentioned in the related works are RL-S2V and ReWatt.
> RL-S2V consider target-attacks, and is of extremely high time costs compared to loss-based methods.
> ReWatt explores graph-level attacks, and only focus on rewiring attacks which are different to most RBA settings.
> So we decide to not include them for the setting differences and efficiency issues following previous loss-based RBAs.

---

> > ### Comment · Reviewer_1uf8 · 2023-11-22
> >
> > I thank authors for their comprehensive reply, it resolved a number of my question. I still have two main concerns regarding the paper:
> > 1. The defender has both structure and features for performing classification task. Indeed in the circumstances when there is extreme heterophily and large feature noise, it is both infeasible for defender and attacker to deal with such data. However, as we lower the degree to which the network shows heterophily, it becomes easier for the defender to perform classification even in the presence of large feature noise (i.e., the classifier learns to ignore the features). But this is not the case with the attacker here as it solely relies on features to perform clustering. Regarding the last part, I don't understand why would we need the assumption that defender cannot perform a protection scheme based on introducing calculated noise to the data and de-noising within the model (which is not accessible to the attacker).
> > 2. If it could be easily incorporated, wouldn't it make sense to include it to have the strongest attacker possible, and then in the experiment section choose whether or not use that feature?

---

> > > ### Author Response · Authors · 2023-11-23
> > >
> > > Thank you for your additional feedback!
> > > 1. In fact, our attack algorithm does not simply lower the homophily / heterophily level of a graph,  but generally create a tougher task targeting all classfiers.
> > > Following [1], both homophiy and heterophily could be beneficial for the defender.
> > > Our attack aims to erase such structural information from the victim graph, where preference of edge formulations become more unclear.
> > > As an RBA, we do not pay much attention to the defender side since we have no access to them.
> > > It is possible to design a good defense model against current RBAs, but it is beyond the scope of the paper.
> > >
> > > 2. The idea of incorporating perturbations to features to yield stronger attacks is interesting, but we do not consider it here for the following reasons:
> > > * General setting in graph adversarial attacks. In graph adversarial attacks, structural attacks attract more attention since they are unique attacks for graph data.
> > > Almost all RBAs and defense baselines mentioned in our paper solely focus on structural attacks.
> > > Besides them, focusing on structural attacks is a common choice, as in [2, 3, 4, 5].
> > > Since we additionally pay attention to attacks under heterophily, we believe it would be better to focus on structural attacks, which does not limit our contribution.
> > > * Ensure fair comparison. While our framework can be easily combined with feature perturbation, the attack result could be unfair compared to previous baselines.
> > > To illustrate the point, we add random noise to the result of SheAttack in Cora-ML with 20% of the perturb ratio.
> > > $X = X + \epsilon * X, \epsilon \sim N(0, \sigma)$.
> > > We control the magnitude ($\sigma$) of the feature perturbation so that the shift of the mean of the norm of each node feature is below 5%
> > > Note that most entries of $X$ are zero so we directly use $\epsilon * X$.
> > > The result is:
> > >
> > > |                           | Evasion | Poison | Avg. Feat Norm |
> > > |----------|----------|----------|----------|
> > > | SheAttack            | 74.72±1.46    | 73.20±0.94      | 1.0  |
> > > | SheAttack(feat)    | 24.68±1.59.   |  57.04±1.21    | 0.98  |
> > >
> > > We see that the attack result is extraordinary when combined with feature perturbation, which makes the comparison to previous baselines meaningless.
> > > * More consideration about feature attacks. The significance of feature attacks could be due to the ill-defined budget for feature perturbations. Here, we found 5% is relatively a large perturbation.
> > >  Also, directly perturbing the features leads to the loss of semantic meaning. The node features are generally not continuous, and a practical feature attack requires far more consideration.
> > > We believe a reasonable evaluation setting must be set up before we dive further into feature attacks on graphs.
> > > As the rebuttal period is ending soon, we are not able to study more feature attacks (e.g, adaptive feature attacks). We would consider including a section discussing feature attacks in the revised version.
> > >
> > > [1] Ma, Yao, et al. Is homophily a necessity for graph neural networks?. In ICLR, 2022.
> > >
> > > [2] Xu, Kaidi, et al. Topology attack and defense for graph neural networks: An optimization perspective. In IJCAI, 2019.
> > >
> > > [3] Geisler, Simon, et al. Robustness of graph neural networks at scale. In NeurIPS, 2021.
> > >
> > > [4] Entezari, Negin, et al. All you need is low (rank): Defending against adversarial attacks on graphs. In WSDM, 2020
> > >
> > > [5] Jin, Wei, et al. Graph structure learning for robust graph neural networks. In SIGKDD, 2020.

---

### Official Review · Reviewer_8KRc · 2023-11-01

**Soundness:** 2 fair
**Presentation:** 2 fair
**Contribution:** 2 fair
**Rating:** 5
**Confidence:** 5

**Summary:**

The authors propose a new black box attack on the graph structure that uses a variant of the silhouette score as one component of the attacker's loss. This captures distances between intra-class/cluster and inter-class/cluster instances and reflects the difficulty of the classification problem. The authors argue that this loss is agnostic to whether the graph is homophilic or heterophilic. Their attack does not require knowledge of the node labels.

**Strengths:**

The theoretical insights are interesting even though they rely on several simplifying assumptions.

The attack can scale to larger graphs such as ogbn-arxiv and ogbn-products.

The black box threat model is relevant and interesting to study but has received relatively less attention in the past.

**Weaknesses:**

The threat model only enforces a global budget, and completely ignores any local constraints e.g. w.r.t. the degree of the nodes. This is likely to lead to unrealistic and noticeable attacks. While the authors perform an empirical analysis in section G and conclude that "unnoticability of SheAttack is in an acceptable range." I do not necessarily agree. First, the averaged results in Table 22 can be misleading since there is likely a big skew in the distribution of changes, and second the mean values are already large.

The experimental evaluation focuses on a fixed perturbation ratio (mostly 0.2 and sometimes 0.1) which can be considered unrealistically large. An in-depth ablation study w.r.t. different perturbation budgets is missing.

The paper would benefit from formalizing and describing the threat model in much more detail. For example, the authors state "only training inputs excluding node labels, are known to attackers." Does this mean that the attacker also does not have access to the training node labels. I assume that this is the case. If yes, a reasonable baseline would be to compare previous (adaptive) attacks [1] using clusters as a surrogate for labels.

If I am wrong and the attacker does have access to training node labels, then they can train a surrogate and use the predictions for the test labels (instead of true labels) which is likely to work much better than using the unsupervised clusters, and likely also better than using "node embeddings generated by supervised GCN as input to generate clusters".

References:
1. Mujkanovic et al. "Are Defenses for Graph Neural Networks Robust?"

**Questions:**

1. How does the attack peform when introducing local budget constraints (e.g. relative to the node degree)?
2. How does the attack compare to an attack where instead of the true labels we use the predictions from the victim model?

---

> ### Author Response · Authors · 2023-11-12
> **Reply to Reviewer 8KRc**
>
> Thanks for your insightful feedbacks!
>
> **Reply to the Weaknesses**
> 1. While perturb ratio above 0.10 might seem unrealistic, a sad fact about RBAs is that it could be extremely hard to find a powerful attack if the perturb ratio is low.
> Take Cora_ml with perturb ratio 0.05 for example, the results against a two-layer GCN are summarized below.
> | Attack  | Evasion | Poison |
> |----------|----------|----------|
> | Clean    | 85.94  | 85.94   |
> | Random | 84.79 | 84.72 |
> | DICE | 84.09 | 83.82 |
> | SPAC-A |84.97 | 84.69 |
> | GFAttack| 84.83 | 83.92 |
> | PEEGA | 82.49 | 81.56 |
> | She | **81.95** | **80.77** |
> | She-A | 82.07 | 80.92 |
>
> While the results of SheAttack is promising, the performance drop is marginal for all RBAs.
> Based on our discussion in Section 7, we highly suspect that the attack performance of white-box attacks would be similarly marginal without utilizing  split information.
> So following the settings of previous RBAs, we set the perturb ratio in most experiments equal of higher than 10%.
> We include experiments with perturb ratio 0.15 in Appendix F.5.
>
> 2. Unnoticability. In our paper, we do not focus on addressing unnoticability for the following reasons.
> * The requirements of unnoticability is varied.
> Besides of the degree shift mentioned, there are homophily shift [1] and spectral shift [2] that might also be noticeable for the defender.
> It is hard for an RBA to tackle unnoticability from all perspective while staying effective.
> *  Difficulties for evaluation. For example, baselines like DICE and PEEGA are based on utilizing homophily shift for performance improvement.
> So it could be meaningless to impose unnoticability constraints about homophily for them.
> Based on our knowledge, it is still hard to ensure a fair evaluation setting with reasonable unnoticability constraint.
>
> Despite the above difficulties, SheAttack could incorporate plug-in unnoticability designs if necessary.
> In the cases where a large degree shift is not allowed, we could include a degree-shift loss into the loss term as a penalty:
> $L_{atk} = L_{she} + \lambda L_{shift} - \gamma L_{deg-shift}$
> We test the performance of She-Comb and She-Comb with degree constraints in Cora-ML with 20% perturb ratio.
> The result is:
> | Attack  | Evasion | Poison | Deg shift |
> |----------|----------|----------|-----------|
> | She-Comb | 74.44±1.95 | 72.06±1.46  | 0.59 |
> | She-Comb (deg-cons) |76.82±1.26 | 75.11±1.07 | 0.01 |
>
> While the attack performance slightly drops, the degree shift is almost eliminated.
>
> 3. Yes, RBAs do not allow the attacker to have access to any node labels.
> For example,  in a social network, the attacker could hack the member accounts, but do not know the downstream tasks of defenders (e.g., whether they predict the genders or regions of members).
> In the experiments, we have include a self-atk attack, which can be seen as GRBCD trained on surrogate labels.
> The attack aims to make the output of the attacked graph different to clusters as labels, but the performance is not satisfactory.
> We will include a more detailed illustration of the RBA setting in the Appendix in the revised version.
>
> **Reply to the Questions**
>
> We have included the reply to questions in the **Reply to the Weaknesses**.
> If you have any further questions, we are willing for more discussion.
>
> [1] Chen, Yongqiang, et al. "Understanding and improving graph injection attack by promoting unnoticeability."  In ICLR 2022.
>
> [2] Lin, Lu, Ethan Blaser, and Hongning Wang. "Graph structural attack by perturbing spectral distance." Proceedings of the 28th ACM SIGKDD Conference on Knowledge Discovery and Data Mining. 2022.

---

### Official Review · Reviewer_e7Jw · 2023-11-02

**Soundness:** 3 good
**Presentation:** 3 good
**Contribution:** 3 good
**Rating:** 6
**Confidence:** 4

**Summary:**

In this paper, the authors study the problem of restrict black-box attacks (RBA) on GNNs. It first introduces the Silhouette Scores, which is used for quantifying the difficulty of a clustering problem, to the RBA attacks on GNNs. Then it introduces a RBA attack named SheAttack by minimizing the silhouette score of the graph. And a scalable version of SheAttack is also proposed for the large-scale graphs. The experimental results on homophily and heterophily graph benchmarks demonstrate its effectiveness compared to other RBA baselines.

**Strengths:**

1. This paper study the problem of restricted black-box attacks, which is both practical and noteworthy.
2. This proposed method is effective in both homophily and heterophily settings. And the scalable version of SheAttack can also work on the large-scale graphs.
3. The experimental results on both homophily and heterophily graphs show that SheAttack can outperform other RBA baselines.

**Weaknesses:**

1. Although the authors include the experimental results on large-scale graphs, the comparison between SheAttack and some existing powerful baselines, such as PRBCD, on the large-scale graphs is missing.
2. In this paper, the authors highlight that SheAttack is applicable to the heterophilic settings while existing RBA methods cannot. However, I think it would be better if some robust GNNs for heterophily graphs can be included during the comparison, such as [1].
3. I recommend the authors can include the comparison of the running time among different methods to verify the efficiency of SheAttack.

[1] Robust Heterogeneous Graph Neural Networks against Adversarial Attacks. AAAI 2022

**Questions:**

1.	Could you please provide some comparisons between PRBCD/PRBCD-shuffle with SheAttack on large-scale graphs?
2.	Please add the experimental results of SheAttack against RobustGNN for heterophily graphs.
3.	Please include the comparison of running time among different methods.

---

> ### Author Response · Authors · 2023-11-12
> **Reply to Reviewer e7Jw**
>
> Thanks for your detailed feedback!
>
> **Reply to the Weakness**:
> 1. PRBCD / GRBCD are white-box attacks that utilize training-test spilts, architecture of victim models, and node labels.
> The results of PRBCD and GRBCD would be extremely strong with these extra information.
> Therefore,  we do not compare our attack with these white-box attacks following previous literature addressing RBAs.
> We include the shuffled version of them that dismiss the splits information in Table 6.
> Without the splits information, PRBCD/GRBCD suffers from severe performance drop even with the knowledge of ground-truth node labels.
>
> We further test PRBCD and PRBCD-shuffle on ogbn-arxiv.
> We see that if the split is shuffled, PRBCD as white-box attacks is just slightly better than SheAttack although it utilizes node labels.
>
> | Attack  | Evasion | Poison |
> |----------|----------|----------|
> | Clean    | 69.60   | 69.60   |
> | PRBCD  | 37.01  | 38.52   |
> | PRBCD-shuffle | 59.06 | 61.61|
>
> 2. We carefully go through the baseline the reviewer mentioned, but it seems that the paper is about heterogenous graphs, which are graphs of multiple edge types.
> The heterophily concept we adopt in the paper is about the preference of connections in edges.
> Nevertheless, we test the performance of SheAttack against a robust heterophilic baseline EvenNet [1] on the synthetic datasets with perturb ratio being 20%.
> The results are summarized below. SheAttack still perform well against the robust defense model.
>
> | Attack      | cSBM+25 Evasion | cSBM+25 Poisoning | cSBM-25 Evasion | cSBM-25 Poisoning |
> |-------------|-----------------|-------------------|-----------------|-------------------|
> | Clean       | 81.95±0.36      | 81.95±0.36        | 80.34±1.15      | 80.34±1.15      |
> | Random    | 81.55±0.61   | 81.26±0.61     | 79.47±0.84   | 79.14±0.93     |
> | DICE         | **79.58±0.64**      | 78.86±1.10       | 83.07±1.08      | 86.78±0.33        |
> | SPEC        | 82.22±0.35      | 81.74±0.33        | 80.00±0.81     | 80.72±0.46        |
> | GFAttack    | 81.68±0.54     | 80.82±1.23        | 79.12±0.41      | 79.41±0.43        |
> | Self-atk       | 81.04±0.28      | 81.01±0.61        | 79.84±1.13      | 79.98±0.58        |
> | PEEGA      | 81.63±0.27      | 81.28±0.63        | 79.74±0.71      | 79.49±0.42      |
> | She            | 79.94±0.33      | **78.69±1.11**   | **78.03±1.07**      | **78.21±0.83**       |
> | She-A        | 79.60±0.38     | 78.72±1.13        | 78.27±0.85      | 77.60±0.99      |
>
> | Attack      | cSBM+50 Evasion | cSBM+50 Poisoning | cSBM-50 Evasion | cSBM-50 Poisoning |
> |-------------|-----------------|-------------------|-----------------|-------------------|
> | Clean       | 91.10±0.37     | 91.10±0.37        | 91.34±0.22      | 91.34±0.22     |
> | Random    | 89.76±0.48   | 89.33±0.54     | 89.02±0.36  | 88.32±0.65     |
> | DICE         | **85.84±0.74**      | **83.66±1.11**       | 95.81±0.39      | 95.78±0.34        |
> | SPEC        | 91.50±0.36      | 91.34±0.45        | 91.65±0.21     | 91.68±0.21        |
> | GFAttack    | 89.90±0.36     | 90.22±0.57       | 89.73±0.56      | 89.18±0.88        |
> | Self-atk       | 90.11±0.27      | 90.13±0.16        | 89.65±0.45      | 89.26±0.56     |
> | PEEGA      |89.26±0.22      | 88.02±0.66        | 89.63±0.52    |88.85±0.68      |
> | She            | 88.58±0.51      | 87.52±1.01       | **86.35±0.63**      | **84.03±0.95**       |
> | She-A        | 89.41±1.01    | 88.67±1.04    | 87.09±1.03      | 85.62±0.88      |
>
> 3. We have included the time efficiency on cSBM datasets and ogbn-products in Appendix F.6.
> If you are interested in the time efficiency of other datasets, we are willing for more discussion.
>
> [1] Lei, Runlin, et al. "Evennet: Ignoring odd-hop neighbors improves robustness of graph neural networks." Advances in Neural Information Processing Systems 35 (2022): 4694-4706.

---

### Official Review · Reviewer_oiQU · 2023-11-06

**Soundness:** 2 fair
**Presentation:** 2 fair
**Contribution:** 3 good
**Rating:** 6
**Confidence:** 3

**Summary:**

This paper focuses on the restricted black-box attack scenario where attackers only have access to node features and the graph structure. To solve this problem, the authors introduce the Modified Silhouette Score (MMS) to measure a graph’s quality and propose a Silhouette score-based attack. Extensive experiments are conducted.

**Strengths:**

1. The studied restricted black-box attack is a practical scenario and an important problem.
2. The motivation of this paper is clear and the idea of introducing the Silhouette Score to measure the quality of a graph is interesting.
3. Extensive experiments are conducted.

**Weaknesses:**

1. The proposed SheAttack depends on various hyper-parameters, which may significantly affect the performance of the proposed method and lack detailed theoretical support or empirical analysis. For example, the number of clusters $k$, the propagation layer/time on the adjacency matrix, and the $\lambda$ that balance the Shift Loss and Silhouette Score-based Loss.
2. The proposed method does not seem to be consistently effective in all scenarios in Tables 1&3.
3. The writing of this paper needs to be further improved. For example, the citation format of references in Introduction and Preliminaries seems strange. 'Aggreation function' should be 'Aggregation function’. The definition of poison attacks and evasion attacks on page 3 is confusing.

**Questions:**

1. In Figure 3, why do GRBCD and RandAttack show different Modified Silhouette Score at epoch 0?
2. How would the proposed method perform when the victim model is not a two-layer GNN? In other words, when the propagation layers/times in the proposed method and victim model are not the same, would the proposed method still be effective?

---

> ### Author Response · Authors · 2023-11-12
> **Reply to reviewer oiQU**
>
> Thank you for your time and efforts!
>
> **Reply to the Weakness**:
> 1. For the hyperparameter $k$, we have included a sensitivity analysis in Appendix F.3. Our results show that our method is not sensitive to it, and we provide suggestions for choosing a proper $k$ in practice.
> For the hyperparameter $\lambda$, we only tune it in within a small range, which is [0, 1.0, 5.0]. the result turned out to be desired enough.
> To sum up, SheAttack is not dependent on well-chosen hyperparameters.
> 2. On synthetic datasets and homophilic datasets, SheAttack is only inferior to DICE in the homophilic setting, which utilizes ground-truth labels and is not strictly an RBA.
> On real-world datasets, SheAttack is inferior to baselines in the poison setting in Chameleon and Squirrel.
> In [1], the authors questioned that the two datasets may face data leakage problems.
> In the filtered Chameleon and Squirrel, the results with 20% the perturb ratio is:
> | Attack      | Chameleon_filtered (Evasion)| Chameleon_filtered (Poison) | Squirrel_filtered (Evasion) | Squirrel_filtered (Poison) |
> |-------------|-----------------|-------------------|-----------------|-------------------|
> | Clean       | 40.36±2.36     | 40.36±2.36 | 35.47±1.00 |  35.47±1.00 |
> | Random    | 35.52±3.44  | 36.50±1.72     | 32.70±2.39 |   35.36±1.55|
> | DICE         | **33.00±2.43**      | **32.91±2.08**     | 28.78±1.38 | 38.81±1.82   |
> | SPEC        | 38.30±2.02      | 37.85±2.43        | 31.22±2.81     | 35.72±0.80      |
> | GFAttack    | 34.44±2.51     | 35.34±4.51       | 32.01±3.10      | 33.88±1.98     |
> | Self-atk       | 39.82±2.48      | 37.13±2.67        | 33.13±3.60    | 35.36±1.64     |
> | PEEGA      | *33.00±2.68*      | 35.96±2.48        | 29.46±2.02    | 33.53±0.50      |
> | She            | 35.78±3.15      | *33.72±2.43*       | **28.38±2.55**      | *33.24±0.93*     |
> | She-A        | 36.23±1.64    | 36.68±1.80    | *28.67±1.39*      | **32.91±2.33**      |
>
> We see that SheAttack holds best performance in the three of four new heterophilic settings.
> A recent study [1] points out that these two datasets also exhibit a mixing pattern of homophily and heterophily, which would be complex for both attackers and defenders.
> As a result, no methods show general superiority in these two datasets.
>
> 3. Thanks you for your advice in the writings. For the citations and writing, we will adopt a unify format and carefully revise the spellings. For the definition about poison attacks and evasion attacks, we only include a brief introduction due to space limitations.
> We adopt the most common definition for them as previous graph adversarial literature, and we will include a detailed explanation to it in the Appendix for the readers' convenience in the revised version.
>
> **Reply to the Questions**:
> 1. In Figure 3, GRBCD and RandAttack exhibits almost the same Modified Silhouette Score at epoch 0 when evaluated given the labels (warm-colored) and clusters (cold-colored).
> The slight difference could be due to visualization issues.
> We would appreciate that if more details about the question could be provided.
> 2. In the experiments against ogbn-arxiv and ogbn-products, we set the number of GNN as 3 following [ogb-example](https://github.com/snap-stanford/ogb/tree/master/examples/nodeproppred/arxiv).
> Based on the result in our paper, the performance of SheAttack is still promising.
>
> [1] Platonov, Oleg, et al. "A critical look at the evaluation of GNNs under heterophily: are we really making progress?." In ICLR 2023.
>
> [2] Mao, Haitao, et al. "Demystifying Structural Disparity in Graph Neural Networks: Can One Size Fit All?." In NIPS 2023.

---

> > ### Comment · Reviewer_oiQU · 2023-11-22
> > **Replys**
> >
> > Thank the authors for addressing some of my concerns. Based on this, I would like to adjust my score accordingly.

---

> > > ### Author Response · Authors · 2023-11-22
> > >
> > > We deeply appreciate your detailed feedback and the re-evaluation of our work.

---

### Author Response · Authors · 2023-11-19
**Summary of the Updates during Rebuttal**

Dear Reviewers, I would like to summarize our updates during the rebuttal period here.

In the Reply to reviewer oiQU:
* We provide new results in heterophilic datasets Chameleon_filtered and Squirrel_filtered, which verifies the superiority of SheAttack in heterophilic settings.

In the Reply to reviewer e7Jw:
* We provide experiment results for PRBCD and PRBCD-shuffle on ogbn-arxiv. PRBCD-shuffle only enjoys a slight advantage over SheAttack while utilizing node labels, which means SheAttack is very close to white-attack without splitting information.
* We conduct attacks in synthetic cSBM datasets against EvenNet, a model that handles heterophily as the defense method. The results show that SheAttack is still powerful against the heterophilic defense method.

In the Reply to reviewer 8KRc:
* We include results in Cora-ML with 5% of the perturb ratio. We did not include it in our paper since 5% is a relatively low perturb ratio for RBAs to achieve a successful attack. Nevertheless, SheAttack continues to hold superiority with a low perturb ratio.
We have included experiment results with 10%, 15%, and 20% in the Appendix. More results with a low perturb ratio will be included in the revised version.
* For the unnoticability, we offer the reasons why SheAttack didn't consider unnoticability in the design. We provide a variant of SheAttack with degree constraints, which eliminate the degree shifts while achieving desirable attack performance.

In the Reply to reviewer 1uf8:
* We address the concerns of the reviewer about the quality of clusters, the choice of $\Delta$, the novelty of our paper, and other questions about model design.

We will carefully go through the writings of our paper. If you have any other questions, we are welcome for further discussion.

Thanks for your time and efforts in reviewing.

---

### Meta-Review · Area_Chair_Fn9E · 2023-12-15

**Metareview:**

This paper proposes a new restricted black-box attack method on graphs. The proposed attack leverages the Silhouette Scores in the attack objective and is claimed to be effective in both homophily and heterophily settings.

The paper studied an important problem, introduced an interesting method, and conducted extensive experiments. However, overall, none of the reviewers are enthusiastic about the proposed attack. Rather, several reviewers raised concerns questioning whether the proposed attack is realistic. During the rebuttal, a few such concerns were not successfully addressed. For example, the proposed attack requires a significant perturbation ratio to be effective. The attack is also not unnoticeable.

While the paper introduces some interesting ideas, the concerns that the attack is unrealistic largely undermine the significance of this work.

**Justification For Why Not Higher Score:**

The attack is deemed unrealistic.

**Justification For Why Not Lower Score:**

N/A

---

### Decision · Program_Chairs · 2024-01-16

Reject